# Compensatory evolution of *Pseudomonas aeruginosa*'s slow growth phenotype suggests mechanisms of adaptation in cystic fibrosis

Ruggero La Rosa [1✉], Elio Rossi [2,5], Adam M. Feist[1,3], Helle Krogh Johansen [1,2,4] & Søren Molin[1]

Long-term infection of the airways of cystic fibrosis patients with *Pseudomonas aeruginosa* is often accompanied by a reduction in bacterial growth rate. This reduction has been hypothesised to increase within-patient fitness and overall persistence of the pathogen. Here, we apply adaptive laboratory evolution to revert the slow growth phenotype of *P. aeruginosa* clinical strains back to a high growth rate. We identify several evolutionary trajectories and mechanisms leading to fast growth caused by transcriptional and mutational changes, which depend on the stage of adaptation of the strain. Return to high growth rate increases antibiotic susceptibility, which is only partially dependent on reversion of mutations or changes in the transcriptional profile of genes known to be linked to antibiotic resistance. We propose that similar mechanisms and evolutionary trajectories, in reverse direction, may be involved in pathogen adaptation and the establishment of chronic infections in the antibiotic-treated airways of cystic fibrosis patients.

[1] The Novo Nordisk Foundation Center for Biosustainability, Technical University of Denmark, Lyngby, Denmark. [2] Department of Clinical Microbiology 9301, Rigshospitalet, Copenhagen, Denmark. [3] Department of Bioengineering, University of California, San Diego, CA, USA. [4] Department of Clinical Medicine, Faculty of Health and Medical Sciences, University of Copenhagen, Copenhagen, Denmark. [5] Present address: Department of Biosciences, Università degli Studi di Milano, Milan, Italy. ✉email: rugros@biosustain.dtu.dk

Cell duplication is a fundamental process in all organisms. However, how adaptive evolution fine-tunes the rate of duplication according to the environmental conditions is still not fully understood. While high growth rate is generally considered a fitness advantage[1,2], in natural environments, reduced growth rate and low cellular activity can instead be favoured or even selected for[3–5]. During the infection of cystic fibrosis patients (CF), *Pseudomonas aeruginosa* may reside in the airways for more than 30 years, broadly modifying its phenotype in response to antibiotics, various stresses and the immune system present in the lungs[6,7]. Growth rates, growth potential and cellular activities are drastically changed as a consequence of adaptive evolution[5,8,9]. The majority of adapted clinical isolates show a slow growth phenotype which is often associated with increased biofilm production and antibiotics resistance, opening the question of the role of reduced growth rate in chronic infections[8]. While slow growth has been described for bacterial pathogens to reduce antibiotic sensitivity and increase bacterial persistence[10–13], it is mostly unknown if in *P. aeruginosa* it is specifically selected for in vivo as a CF specific trait, or if it is a mere consequence of other adaptive mutations with high fitness cost such as mucoidy, antibiotic resistance and biofilm production. Moreover, the molecular causes of the reduced growth rate are entirely uncharacterized in clinical isolates. Nevertheless, reduced growth rate is an important phenotypic marker of evolution progression from naïve/environmental strain to adapted/clinical strain, which could be used diagnostically to improve treatment of the patients through more efficient eradication of the infection.

Convergent evolution has previously been described in *P. aeruginosa* clinical isolates and convergent phenotypes have also been characterized[8,14–17]. Interestingly, the dynamic CF airways environment can select for bacterial variants with increased mutation frequency (hypermutators), which can quickly reach a new fitness peak[18–20], favouring the acquisition of beneficial traits such as enhanced antibiotic resistance, virulence, persistence and metabolic adaptation in a much shorter time[21]. Hypermutator subclones can represent up to 60% of *P. aeruginosa* population in CF airways and 30–50% of the infected CF patients have hypermutator isolates, indicating a critical role of hypermutation during within-host adaptation[22–28].

So far, to study adaptive evolution two main approaches have been used: on one hand the advance in whole genome sequencing (WGS) has made it possible to analyze longitudinal collections of bacterial isolates, to identify trajectories of evolution and, in some cases, to correlate mutations to phenotypes[8,14,15,17,25]; on the other hand, adaptive laboratory evolution (ALE) has proven useful to recreate evolutionary processes in vitro and to compare them to the ones occurring in natural environments[29,30]. However, several distinct genomic solutions can lead to the same phenotype, hampering the identification of genetic markers of evolution. Moreover, selecting for a phenotype such as "slow growth" in vitro is challenging because the specific selective pressures enriching for reduced growth rate are still undetermined.

To overcome these limitations, we employed ALE to revert the slow growth phenotype of clinical strains of *P. aeruginosa* and to map adaptive trajectories to high growth rate which, in reverse direction, may have caused the reduced growth rate in CF patients. We investigated the degree of evolvability, the genetic basis of the increased growth rate, and its effect on antibiotic susceptibility of three clinical isolates of *P. aeruginosa* at intermediate and adapted stages of evolution. To select for the highest fitness, we took advantage of the enhanced evolvability of hypermutator strains and their role in adaptation to the CF environment, and evolved these clinical strains in rich medium in

absence of selective pressures. Using this approach, we recreated a sudden change in selective pressures, like those occurring in the CF environment allowing for a rapid selection of strains with the highest fitness in the new conditions. Our analysis reveals that the reversion of the slow growth phenotype may be achieved in steps. First, by transcriptional remodelling and subsequently by accumulation of mutations in global regulators and in genes encoding essential components of the cell. Moreover, increased growth rate is accompanied by increased sensitivity to distinct antibiotics, underlining the trade-off between antibiotic susceptibility and growth rate, which is often neglected. Altogether, these results suggest evolutionary processes occurring during evolution of *P. aeruginosa* in CF patients.

## Results

**Fitness trajectories of evolving *P. aeruginosa* clinical isolates.** To test whether clinical isolates of *P. aeruginosa* after years of within-patient evolution retain their evolvability, we employed adaptive laboratory evolution (ALE) in absence of selective pressures. We evolved, along with the laboratory strain PAO1, three representative hypermutator clinical strains, isolated from three distinct CF patients, with independent evolutionary histories, representing intermediate (strains 141_S and 10_S) and adapted (strain 427.1_S) stages of within-patient evolution[8]. We selected hypermutator strains for their enhanced evolvability and because they represent a major fraction of the *P. aeruginosa* isolates from CF patients[21,22]. Indeed, from the re-analysis of our collection of 474 clinical strains isolated from young CF patients[14], more that 50% of the patients have a first isolate with mutations in *mutS* or *mutL*, representing 30% of the identified clone types (lineages differing by more than 10,000 SNPs) (Supplementary Fig. 1a). The strains were isolated 1, 5 and 27 (respectively strain 10_S, 141_S and 427.1_S) years after the first *P. aeruginosa* was identified in each of the patients at the Copenhagen CF Clinic[14]. Strains 10_S and 141_S belong to relatively new clone types (DK06 and DK15) identified in young children, while strain 427.1_S belongs to an old clone type (DK01) with a long evolutionary history widely spread among old chronically infected patients in Denmark[14,31]. Each starting strain (S) was evolved in three independent parallel ALE experiments in LB medium for maximizing the increase in growth rate. To show the average trajectories of evolution, the data of the parallel experiments from each S strain were merged in combined populations (CP).

The average growth rate (cell doublings per hour ± SD) of the S strains was $0.37\,h^{-1} \pm 0.03$, and after 885 (141_CP), 776 (10_CP) and 913 (427.1_CP) generations of growth, it increased to $1.28 \pm 0.02$, $1.08 \pm 0.02$ and $0.87 \pm 0.02\,h^{-1}$ for 141_CP, 10_CP and 427.1_CP, respectively (Fig. 1a). PAO1 maintained a stable growth rate during the entire experiment (Fig. 1a and Supplementary Fig. 6a). As reference, other clinical strains belonging to the same clone types show a high growth rate indicating that the evolved strains took advantage of their adaptive potential reaching fitness peaks similar or in the direction of their ancestors[8,31].

To examine and predict the fitness trajectories of the evolving strains, i.e., the increase in growth rate, we employed a power-law model which describes the evolvability of the strains as a function of the growth rate and the elapsed generations. The same model was previously used to explain observations of long-term evolution experiments in *Escherichia coli*, *Saccharomyces cerevisiae* and *Schizosaccharomyces pombe*[32–34]. The model does not have an upper asymptote but predicts a slowdown in the rate of evolution as the population increases the fitness. The estimations from the models describe with good accuracy (NRMSE between 0.0852 and 0.1669) the observed increases in growth rate for both

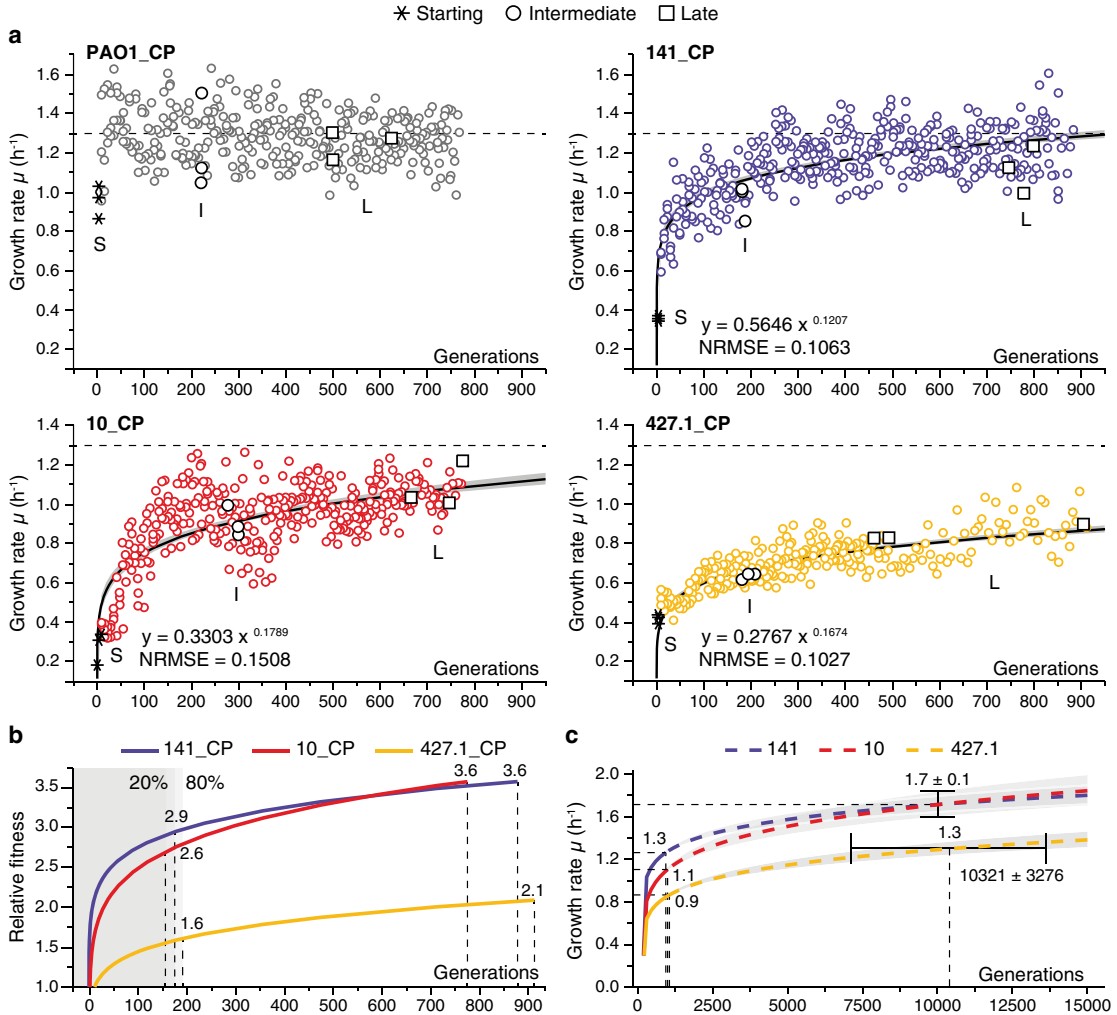

**Fig. 1 Fitness trajectories of evolving *Pseudomonas aeruginosa* populations. a** The plots represent the increase in growth rate of the combined populations (CP) constituted by merging the data of three independent populations evolved in parallel from each starting strain PAO1_S, 141_S, 10_S and 427.1_S (see Supplementary Fig. 2 for each independent population). The dots represent the growth rate of each population during ALE. The asterisks represent the growth rate of each starting (S) strain. The black circles and squares indicate the populations from which single colonies were isolated and further analyzed. The black curves show the fit of a power-law model and the grey shaded area the 95% confidence limits. The models, including the normalized root-mean-square error (NRMSE), are indicated in each plot. **b** Relative fitness observed in the CP. The average fitness increase of each CP was calculated relative to the growth rate of the starting S strains using the power-law model. The fitness at 20 and 80% of generations is indicated in the plot. **c** Growth rate increase prediction for the evolving populations. The continuous lines show the fitness trajectories determined using the experimental data as in panel **a**, while the dashed lines show the simulated posterior distribution of the growth rate according to the power-law models for each CP. The grey shaded areas and black bars show the 95% confidence limits of the predictions. Source data are provided as a Source Data file.

CP and for each single population (Fig. 1a and Supplementary Fig. 2), indicating similar trajectories of evolution between the strains. The fitness relative to the S strain increased by 2.9-, 2.6- and 1.6-fold for CP 141, 10 and 427.1, respectively, during the first 20% of generations, during which most of the fitness increase takes place, and increased by 3.6-fold for both CP 141 and 10 and 2.1-fold for CP 427.1 at the end of the ALE (Fig. 1b).

Since population 427.1_CP showed lower fitness increase relative to the other CP and to other clinical strains of the same clone type[31], we used the power-law model to predict the time needed by the CP to reach the same growth rate as that of PAO1_S (Fig. 1a). First, we validated the robustness of the model in predicting the fitness trajectories (Supplementary Fig. 3; see the Methods section for a full description). Next, we predicted that, if the same trajectory of evolution is maintained, population 427.1_CP would match PAO1 growth rate after 10,321 ± 3,276 generations (Fig. 1c). To further confirm the reliability of the

model, we predicted a growth rate of 1.7 ± 0.1 h⁻¹ at 10,000 generations for CP 141 and 10, which is a reasonable growth rate for *P. aeruginosa* in optimal conditions.

To evaluate whether the ALE condition selected specifically for increased growth rate rather than other phenotypes, we isolated two random representative strains from each single population at an intermediate (I) and late (L) time point of ALE (Fig. 1a) and evaluated changes in phenotypes known to be affected during ALE experiments performed in the same conditions used in our work (Supplementary Fig. 4a). Pearson's correlation analysis of growth rate relative to motility, biofilm and pyoverdine production, revealed no significant correlation ($r = 0.50$, $−0.08$ and $−0.09$) between the variables (Supplementary Fig. 4b). This indicates that our conditions imposed a strong selective pressure on growth rate rather than selecting for other phenotypes.

These results show that the clinical isolates maintained their evolvability, which is indispensable for adaptive evolution.

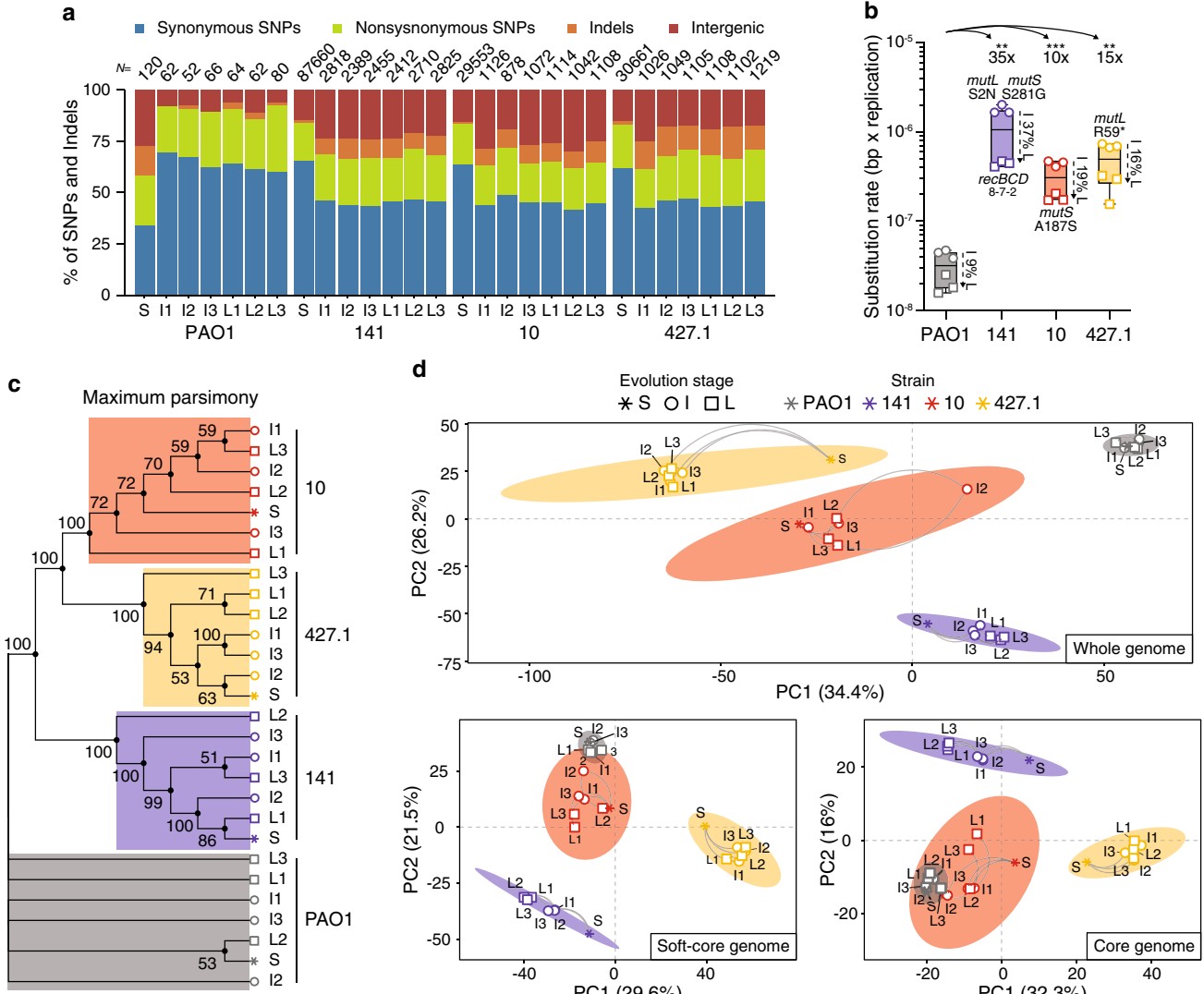

**Fig. 2 Mutational pattern and transcriptional landscape of evolving *Pseudomonas aeruginosa* strains. a** Distribution of the different types of mutations identified in the genomes of the starting (S), intermediate (I) and late (L) strains. The total number of mutations identified relative to PAO1 (S strains) or relative to each S strain (I and L strains) are shown above each column. **b** Substitution rate (bp per generation) for I and L evolved strains. Differences between strains were calculated on the average of the substitution rate for I and L strains. The fold-change and the significance (two-tailed Student *t*-test, *P* values: **<0.0021; ***<0.0002; *n* = 6 biological independent strains) relative to PAO1 are shown. The decrease in substitution rate observed in L strains is calculated as percentage relative to I strains. Data are presented as minimum, 25th percentile, median, 75th percentile, and maximum. Amino acid changes likely causing hypermutability are indicated for each strain. Numbers under the *recBCD* genes (141 strains) indicate the number of nonsynonymous mutations identified in each gene. **c** Maximum parsimony reconstruction of the sequenced strains. The tree is based on 8065 missense and nonsense SNP mutations accumulated during ALE. Branches corresponding to partitions reproduced in less than 50% of bootstrap replicates are collapsed. The percentages of replicate trees in which the associated taxa clustered together in the bootstrap test (500 replicates) are shown next to the branches. **d** Unsupervised principal component (PCA) analysis performed on whole, soft-core and core genome expression data for each strain. The ellipses represent the k-means clustering of the transcriptional profiles of the strains. Source data are provided as a Source Data file.

However, the historical contingency, i.e., the specific combination of mutations previously accumulated during within-patient evolution, influenced the rate and level of fitness increase and thus the ability to adapt to a new environment.

**Shifts in the mutational pattern and in the transcriptional landscape during evolution.** To gain insight into the physiological events that could explain the reversion of the growth phenotype, we compared the clinical S strains to the PAO1 reference strain and the I and L evolved strains to the corresponding S strains using comparative genomics and transcriptomics (Figs. 1a and 2; Supplementary Data 1–6). Using this approach, we identified mutations in S strains relative to the reference genome of

PAO1 and mutations in I and L strains relative to each S ancestor. In strains 141_S, 10_S and 427.1_S we identified 87,660, 29,553 and 30,661 mutations, respectively, relative to the reference genome of PAO1, while in the I and L strains, we identified on average 1587 ± 750 between newly accumulated mutations and reversions of mutations already present in the S strain (Fig. 2a).

Due to the presence of nonsynonymous mutations in the DNA mismatch repair system genes *mutL* and *mutS* and in the recombinational DNA repair system *recBCD* in the S strain genomes (Fig. 2b; Supplementary Data 1), the substitution rates, determined as bp per genome per replication, were between 10- to 35-fold higher than for PAO1 (Fig. 2b). The substitution rates for the I and L strains decreased on average 20% (9% for PAO1,

37% for 141, 19% for 10 and 16% for 427.1), which fits well with the slow-down in the rate of evolution as also described by the power-law model (Figs. 1b and 2b). The dN/dS ratio ($0.16 \pm 0.02$) was comparable in all strains and implied negative selection (probability of neutral selection <0.0001). The difference in substitution rate between populations can only partially explain the distinct evolvability of the strains, since strain 10 and 427.1 show comparable substitution rates but different fitness trajectories (Figs. 1a and 2b).

Both maximum parsimony and maximum likelihood phylogenetic analysis of the concatenated SNP mutations show that each strain evolved independently but remains clustered according to its ancestor overall maintaining the linear relationship between S, I and L strains (Fig. 2c, Supplementary Fig. 5). Similarly, an unsupervised principal component (PCA) analysis and k-means clustering performed on whole-genome expression showed that strains originating from the same ancestor remain clustered together despite their respective distribution along the PC1 axis (34% variance explained) (Fig. 2d). Presence or absence of genes due to long-term adaptation can strongly influence PCA analysis[35–37]. Therefore, we also evaluated the overall similarities between the transcriptomes covering genes belonging to the species soft-core ($n = 5037$, 88%) and core genome ($n = 2319$, 40%)[35]. Removing accessory genes strongly reduces the bias introduced by genetic difference and the two restricted sets are enriched with core functionality, i.e., central carbon metabolism and respiratory genes, that are more likely to contribute to the growth potential (Fig. 2d). Evolved strains derived from PAO1, whose growth phenotypes already represent an optimum in the fitness landscape, showed a reduced variability and clustered in the same region as the S strain. The strains able to achieve growth rates similar to the reference strain PAO1 (10 and 141), followed a trajectory on the PC1 axis directed towards the area occupied by the PAO1 strains, suggesting that global physiological changes affecting the core genome are responsible for the change in growth rate (Fig. 2d). In contrast, strains originating from 427.1_S, which do not fully recover the growth phenotype, distributed along the PC1 axis in the opposite direction. This indicates that mutations accumulated in this lineage constrain the core genome expression, preventing a full recovery of the growth phenotype and directing the strain towards another, probably sub-optimal, fitness peak. Independent of directionality, replicates from each S strain followed similar evolutionary trajectories indicating strong parallel evolution (Fig. 2d).

Collectively, the RNA-seq data confirmed what was suggested by the WGS analysis: the different strains evolved individually (displaying parallel evolution) but remained clustered to their respective ancestors, probably due to the strong mutational signature of the starting strain. Further, the analyses of the overall gene expression profiles suggest that conserved components of cell physiology rooted in the core genome are to a large extent responsible for the observed phenotypes.

**Reversion of the slow growth phenotype**. In an attempt to correlate reduced growth rate of the clinical strains with specific mutations or with specific changes in their transcriptional profiles, we investigated which genes were potentially involved in increasing growth rate in ALE conditions according to the following criteria. Due to the high number of accumulated mutations, we filtered out: (1) synonymous mutations; (2) mutations in intergenic regions; (3) mutations in genes already containing more than 3 non-synonymous mutations in the S strain, which might indicate loss-of-function in the CF environment; (4) mutations in genes coding for hypothetical and uncharacterized proteins. Using this approach, we reduced the noise caused by

hypermutability, and identified 17 (strains 141), 8 (strains 10) and 91 (strains 427.1) genes potentially involved in causing the increased growth rate (Fig. 3; Supplementary Data 1). Here, only the specific genetic changes correlating with an increase in growth rate will be presented in detail. An overview of the mutations, changes in the transcriptional profiles and of the growth rates of the analyzed strains can be found in Supplementary Fig. 6 and Supplementary Data 1–6. For simplicity each strain will be described separately.

**Alginate metabolic burden in strains 141**. In the isolates originating from strain 141_S, expression of the alginate regulators *algU* and *mucA* genes was significantly reduced, both in I and L strains relative to the S strain, and as a direct consequence more than 20% of the genes in the *algU* regulon were downregulated (Fig. 3a, Supplementary Fig. 7a; Supplementary Data 3–4). Comparing the transcription profiles for the 141 and PAO1 strains showed that the difference in the expression level of the *algU* and *mucA* genes between S strains was reduced in L strains to a level similar to that determined for PAO1 (Fig. 3b). However, no mutations in the *algU* and *mucA* genes or promoters, nor in any other known factors involved in their regulation, can explain the reduced expression, suggesting that epistatic effects may reduce *algU* and *mucA* expression to the PAO1 level.

In order to test the effect of *algU* overexpression, a plasmid containing an arabinose inducible *algU* gene was introduced in strains I1, L1 and PAO1 as control. The growth rate of these transformed strains was strongly reduced in presence of 0.5% of arabinose, confirming that increased expression of the *algU* gene significantly reduces growth rate, possibly due to an increased metabolic burden or a redirection of sugar metabolism from energy production to EPS biosynthesis (Fig. 3c, Supplementary Fig. 8a). Not surprisingly, many clinical strains harbour a mutated *algU* gene indicating that, long term, alginate overproduction has a disproportional fitness cost for the cell (Supplementary Fig. 1c).

**DNA topology and nitrogen metabolism in strains 10**. In the strains derived from strain 10_S, 10_I1 and 10_I3, two distinct mutations in the gene encoding the DNA topoisomerase I gene, *topA*, were identified (Supplementary Figs. 6c, 9; Supplementary Data 1). DNA topoisomerases together with DNA gyrases regulate the degree of supercoiling of DNA, hence influencing important processes such as replication and transcription[38]. Loss of DNA topoisomerase I in *E. coli* has a deleterious effect, which can be compensated by mutations in the gyrase genes, *gyrA* and *gyrB*[39]. Interestingly, the 10_S strain harbours a 6-nucleotides deletion in the *gyrA* gene, which might be compensated in strains I1 and I3 by two distinct missense SNPs in the *topA* gene (Supplementary Fig. 9). Since these mutations are maintained in the L strains, we hypothesize that the genetic changes in the *topA* gene may be responsible for the observed growth rate increases (~2-fold) in strains I1 and I3 (Supplementary Fig. 6c). The additional increase in growth rate to the maximum of ~1 $h^{-1}$ (Supplementary Fig. 6c) may be caused by the subsequent indel mutations identified in the *rpoN* sigma factor in the L1 and L3 strains. The S strain harbours a deletion of a T residue at position 1082 in the *rpoN* gene which causes a premature stop of the translation (Supplementary Fig. 9). Indeed, when comparing gene expression in strains 10_S and PAO1_S (Supplementary Data 2) an enrichment of RpoN-dependent genes in the repressed set was observed (Fig. 3d, Supplementary Fig. 7b). The subsequent indel mutations (1079insT and 1090insG), located in the proximity of the 1082delT deletion, re-establish the reading frame of *rpoN*, ensuring the complete translation of the protein (Supplementary Figs. 6c and 9). Accordingly, expression of the

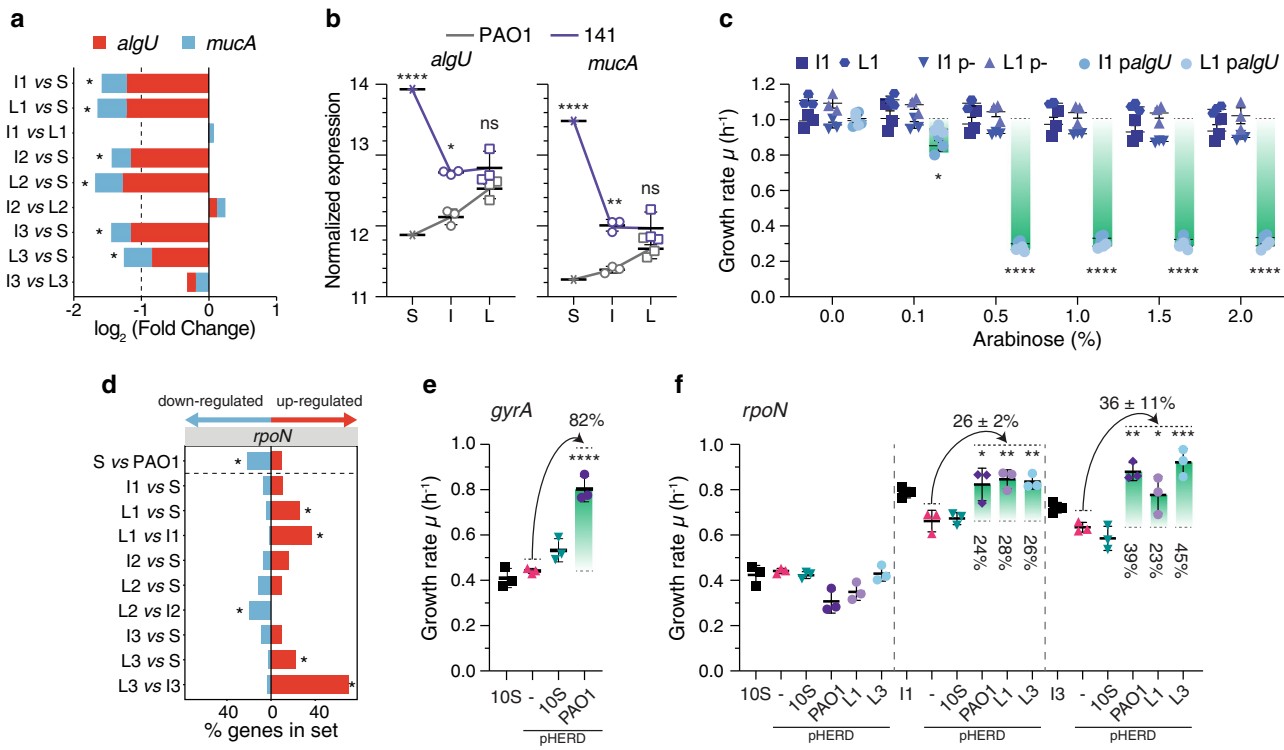

**Fig. 3 Mutations and transcriptional profile changes leading to increased growth rate in the evolved strains. a** Expression of *algU* and *mucA* genes in 141 strains. The asterisks indicate statistically significant differences in the level of expression for both genes among the compared pairs of strains calculated by negative binomial distribution testing (adjusted *P* value < 0.01). **b** Normalized expression level of *algU* and *mucA* genes in S, I and L strains of PAO1 and 141. Statistical significance was calculated by two-way ANOVA with Sidak's multiple comparison test (*P* values: *<0.05; ****<0.0001; ns not significant; *n* = 3 biologically independent strains). Data are presented as mean values ± SD. **c** Growth rate reduction in strains 141_I1 and 141_L1 upon expression of *algU* gene from plasmids p- (empty plasmid) and p*algU* (*algU* gene from strain 141_S) at increasing concentration of arabinose. Statistical significance relative to the absence of inducer (0%) was calculated by two-way ANOVA with Dunnett's multiple comparison test (*P* values: *<0.05; ****<0.0001; *n* = 3 biologically independent experiments). Data are presented as mean values ± SD. **d** Distribution of differentially expressed genes within the *rpoN* sigma factor regulon in strains 10. Genes within the *rpoN* regulon were obtained from refs. [63,71]. The percentage of genes up- and down-regulated is reported in each pairwise comparison. Asterisks denote sets of genes significantly enriched (adjusted *P* value ≤ 0.05, hypergeometric test after Bonferroni correction). **e, f** Complementation of *gyrA* and *rpoN* mutations in strains 10. **e** Growth rate changes in strain 10_S upon expression of *gyrA* gene from plasmids pHERD: - (empty plasmid), 10_S (*gyrA* gene from strain 10_S) and PAO1 (*gyrA* gene from strain PAO1). **f** Growth rate changes in strains 10_S, 10_I1 and 10_I3 upon expression of *rpoN* gene from plasmids pHERD: - (empty plasmid), 10_S (*rpoN* gene from strain 10_S), PAO1 (*rpoN* gene from strain PAO1), L1 (*rpoN* gene from strain 10_L1) and L3 (*rpoN* gene from strain 10_L3). For complementation assays, 0.1% of arabinose was used. Growth rate differences were calculated relative to the strain expressing the pHERD(-) empty plasmid by one-way ANOVA with Dunnett's multiple comparison test (*P* values: *<0.05; **<0.0021; ***<0.0002; ****<0.0001; *n* = 3 biologically independent experiments). If not otherwise indicated, differences between strains were not significant. Data are presented as mean values ± SD. Source data are provided as a Source Data file.

RpoN-dependent genes was significantly upregulated in strains L1 and L3 when compared with the S, I1 and I3 strains (Fig. 3d, Supplementary Fig. 7b; Supplementary Data 5). Surprisingly, more than 40% (*gyrA*) and 20% (*rpoN*) of the clone types have mutated first isolates, suggesting that such mutations might have a functional role also in the environment (Supplementary Fig. 1b).

In order to test whether the mutations in *gyrA* and *rpoN* are involved in the reduction of the growth rate, complementation tests were performed: (1) strain 10_S with the mutant copy of *gyrA* gene from strain 10_S and with the wild-type copy from PAO1; (2) strains 10_S, 10_I1 and 10_I3 with the mutant copy of *rpoN* from strain 10_S, the wild-type copy from PAO1 and copies from strains 10_L1 and 10_L3 containing the compensatory mutations; (3) strain PAO1 with all the same *gyrA* and *rpoN* variants as a control. The complementation of the *gyrA* mutation was only achieved in presence of the wild-type copy from PAO1 (pHERD_PAO1 plasmid), which resulted in an 82% increase in growth rate relative to the strain containing the empty vector plasmid pHERD (-) (Fig. 3e). Complementation of the *rpoN*

mutation was not achieved in strain 10_S, since appropriate expression of the *rpoN* regulon may depend on DNA accessibility, which is achieved only in connection with the additional *topA* mutations in the I1 and I3 strains (Fig. 3f). Accordingly, complementation of the *rpoN* mutation in the I1 and I3 strains showed that the PAO1, L1 and L3 copies of *rpoN* can increase growth rate by 26 and 36%, respectively, relative to strains with the empty vector plasmid pHERD (-) (Fig. 3f). It is worth noting that the empty vector plasmid pHERD (-) has a physiological cost in strains 10_I1 and 10_L1, slightly decreasing growth rate in presence of the inducer. Complementation of both *rpoN* and *gyrA* genes was achieved even in absence of inducer indicating that a very low amount of RpoN and GyrA proteins, synthesized due to a low basal level of expression, is sufficient to restore the growth of the strains (Supplementary Fig. 8b, d). A comparable reduction of growth rate was also shown by a PAO1 derivative strain (*rpoN*-KO) lacking *rpoN* gene which showed 56% lower growth rate relative to the PAO1, confirming the role of *rpoN* for the optimal physiology of the cell even in a clean background (Supplementary Fig. 8f). Overexpression of the *gyrA* and *rpoN* variants in PAO1

had no effect underlining the importance of the genetic context for the functional complementation of the phenotype (Supplementary Fig. 8c, e). In the strains derived from population 2 (I2 and L2), we hypothesize that a missense SNP in the *ntrC* gene might have caused a full recovery of the growth rate (Supplementary Fig. 6c). NtrC is an enhancer-binding protein that activates the transcription of the σ54-holoenzyme by locally modifying the topology of the DNA. Therefore, the *ntrC* mutation might combine the required change in DNA topology with the partial activation of the *rpoN* regulon.

**Compensatory mutations of central component of the cell in strains 427.1**. Due to the large number of genes targeted by mutations, we performed overrepresentation test using Gene Ontology terms to identify categories of gene enriched in strains deriving from the 427.1_S clone. The analysis showed that mutations in genes involved in ethanol oxidation (GO:0006069), DNA replication (GO:0006260), phosphorylation (GO:0016310) and signal transduction (GO:0007165) categories were enriched 37-fold, 8-fold, 3-fold and 3-fold, respectively, (Binomial test with False Discovery Rate correction; FDR < 0.05). In the strains I and L isolated from the 427.1 populations, non-synonymous mutations in genes *lig* (DNA ligase), *dnaG* (DNA primase) and *holC* (chi subunit of the DNA polymerase III), all involved in DNA replication, were observed; these mutations possibly compensate for the non-synonymous SNPs mutations already present in the S strain (Supplementary Fig. 6d). Similarly, strain 427.1_L3 accumulated a missense mutation in *rpoB* (DNA-directed RNA polymerase beta chain) to compensate for 3 missense SNPs in the S strain. In strain 427.1_L1, there is a missense mutation in the *rplU* gene encoding the bL21 protein of the ribosome. Moreover, searching for other mutations among ribosomal genes revealed that all 427.1 strains harbour missense mutations in genes coding for the uL3, uS2, bS16 and uS19 ribosomal proteins and in the gene encoding elongation factor G, confirming that the ribosomal machinery is under selective pressure in the CF environment[40]. In strains L1 and L2, we also found mutations in the *relA* gene (stringent response control system), which may compensate for a mutation in the *spoT* gene in strain 427.1_S (Supplementary Fig. 6d). Not surprisingly, many central components of the cell are under high selective pressure in the CF environment, since tuning the cellular physiology to the host environment is required for a persistent infection (Supplementary Fig. 1c).

**Convergent evolution from CF to the ALE environment**. Although the adaptive trajectories of the tested *P. aeruginosa* strains were found to be quite diverse, strong signs of convergence between adaptive mechanisms were also observed at the genetic and transcriptional levels.

In the evolved strains within the filtered genes, we identified mutations in genes involved in (1) flagellum biogenesis (*flgG* and *fliD*), (2) adaptation to the ALE substrates (*hisP*, *hutH*, *oprB* and *morA*), (3) in oxygen respiration (*cioA* and *cioB*), (4) in biofilm and alginate production (*algW* and *sspA*), (5) in the type 3 secretion system (*exsA* and *spurG*) and (6) pyoverdine production (*fpvI*). These factors are dispensable in liquid and outside the host environment and were also mutated in PAO1 strains (Supplementary Fig. 6; Supplementary data 1).

When compared with PAO1, the 141_S and 427.1_S strains showed a marked increase in the expression of the AlgU regulon with a significant enrichment of genes controlled by the transcriptional regulator in the group of upregulated genes (Supplementary Fig. 7a). During the subsequent ALE experiments, a reversion of this pattern was observed in both lineages, for which expression of AlgU-dependent genes were consequently

enriched in the downregulated fraction (Supplementary Fig. 7a). A strong shift in the transcriptional pattern of the regulon occurred in the descending evolved strains derived from 141_S as denoted by hierarchical clustering (Fig. 4a). Interestingly, the levels of gene expression were reduced to levels close to what was found for PAO1, the most fit strain in our experimental conditions (Fig. 4a).

A similar pattern of gene expression could be observed for the RpoN regulon. All ALE evolved lineages modulated their RpoN-dependent transcriptional profiles to closely match the gene expression levels of the PAO1 strain, thus reverting the different consequences of long-term infection within the RpoN regulon described for the S clones (Supplementary Fig. 7b). This modulation seems to be independent of mutations in the RpoN regulator, since mutations in this gene could only be identified in the strains of the 10 lineage. Hierarchical clustering analysis based on expression of RpoN-dependent genes shows that L strains from both 141_S and 10_S strains cluster together with PAO1 strains, far from the other strains (Fig. 4b). The *nos*, *nor* and *nir* operons involved in the transformation of $NO_2$ in $N_2$, directly regulated by the DNR and NirQ regulators, but under the global control of RpoN, increased their expression in all but the 427.1 strains to the level determined for PAO1 (Fig. 4c). Interestingly, expression of these genes is highly repressed in the CF sputum, probably to avoid toxic effects from NO produced during anaerobic respiration, a condition which is not occurring in the ALE condition[41].

Finally, genes involved in the acute (T3S and type 4 pili) and chronic (T6S, EPS, QS and diguanylate cyclase) phenotypes switched their expression profile due to the accumulation in all L strains of non-synonymous mutations in the *gacS*, *gacA* and *rsmA* genes, part of the regulatory system Gac/Rsm[42] (Fig. 4d, e; Supplementary Data 1). Using hierarchical clustering analysis for expression levels of genes belonging to the Gac/Rsm regulon, it was observed that L strains of 141 and 10, which show higher expression of genes involved in acute infection (high T3S and T4P, low T6SS and EPS; Fig. 4d), clustered far from the respective I and S strains (Fig. 4f), characterized by transcriptional patterns typical for chronic infections (low T3S and T4P, high T6SS and EPS) (Fig. 4d–f). This switch in phenotype is also in line by the increased motility of L strains of 141 and 10 relative to the S strains (Supplementary Fig. 4a). However, even though a switch in the activity of Gac/Rsm-dependent genes during ALE was observed, many genes involved in virulence factor production accumulated non-synonymous mutations, both during within-patient evolution (S strains) and during ALE (I and L strains) (Fig. 4g). This observation suggests that, although adaptive evolution targeting the Gac/Rsm system foster the reactivation of acute-like traits, actual virulence of these strains was not fully reconstituted in the ALE experimental conditions.

These results suggest that, in the CF environment, convergent evolution of CF-specific traits may occur by transcriptional changes induced by tailored activities of global regulators, independent of the accumulated mutations. Indeed, the reversion of the same regulatory mechanisms during adaptation to the ALE occurs independent of patient and specific lineage.

**Increased growth rate leads to increased antibiotic susceptibility**. Withstanding antibiotic treatments is often an energetically expensive trait for bacteria, which may be alleviated by compensatory mutations and changes in transcriptional patterns, or lost as soon as the selective pressure is removed[43].

To test whether antibiotic susceptibility profiles are affected in our ALE condition, we analyzed the minimum inhibitory concentrations (MIC) for six clinically relevant antibiotics. While

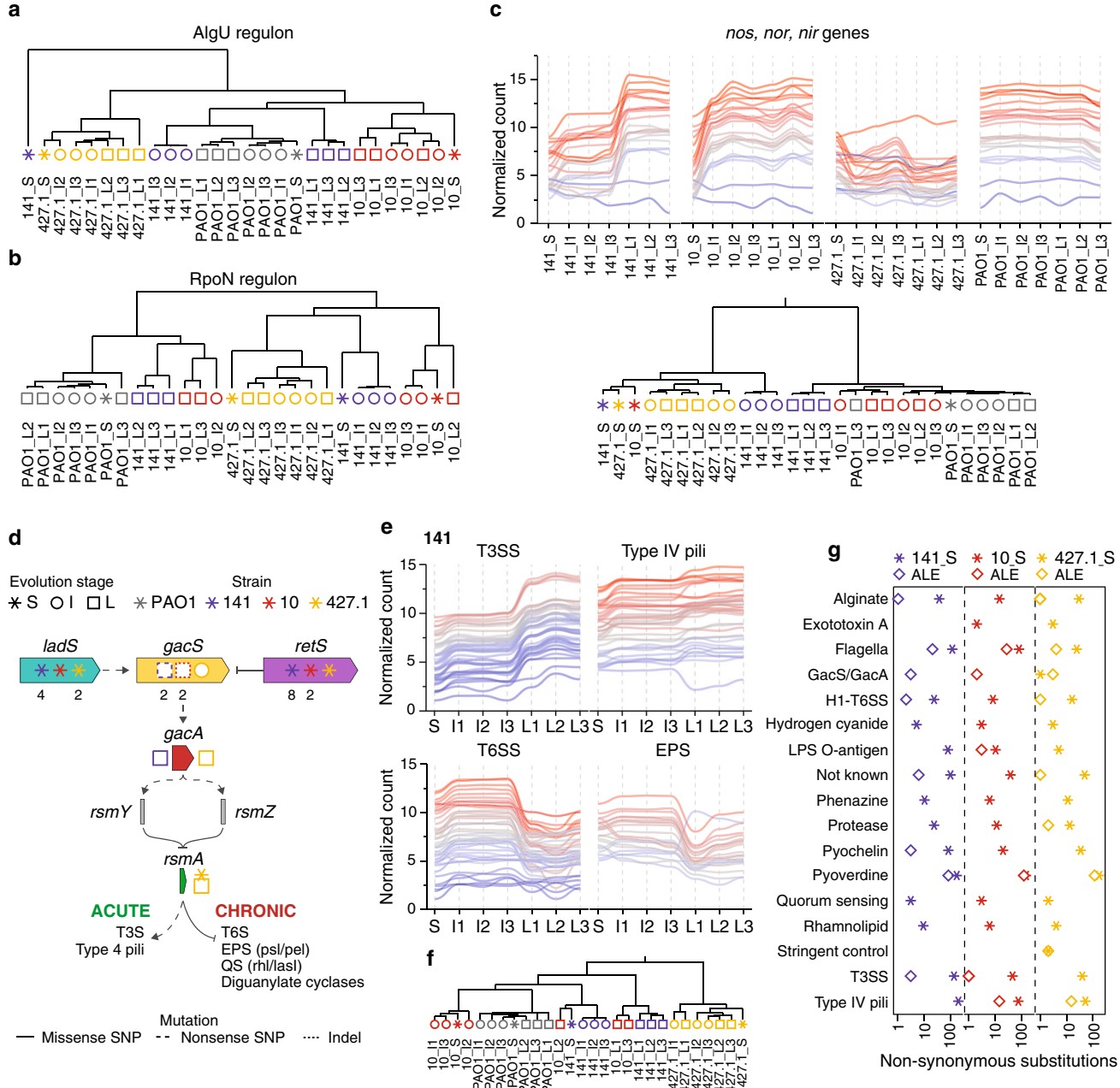

**Fig. 4 Convergent mechanisms of adaptation to the ALE condition.** Hierarchical clustering analysis of the evolved strains based on the expression profile of genes within the **a** *algU* and **b** *rpoN* regulons. Hierarchical clustering was inferred on the Spearman correlation using the Euclidean distance. **c** Normalized expression profile of *nos*, *nor* and *nir* genes in the evolved strains and clustering analysis inferred on the Spearman distance using the complete linkage method. **d** Schematic representation of the Gac/Rsm regulatory network along with the mutations present in the starting (S), intermediate (I) and late (L) strains. Numbers under the symbols indicate the number of mutations identified in the genes. **e** Normalized expression profiles of type 3 secretion system (T3SS), type IV pili, type 6 secretion system (T6SS) and exopolysaccharides *psl* and *pel* (EPS) genes regulated by the Gac/Rsm regulatory network in strains 141. **f** Hierarchical clustering analysis of the evolved strains based on the expression level of acute (T3SS and type 4 pili) and chronic (T6SS, EPS, quorum sensing (QS) and diguanylate cyclases) genes phenotypes. Clustering was inferred on the Spearman distance using the complete linkage method. **g** Number of non-synonymous substitutions (missense and nonsense SNPs and indel mutations) in virulence factor genes identified in the S strains (asterisk) or accumulated during the ALE (diamond). The categories were obtained from the Victors[68], VFDB[69] and PseudoCAP[70] databases. Source data are provided as a Source Data file.

PAO1 and 141 strains showed no (PAO1) or little (141) changes in the MIC values, in strains derived from 10 and 427.1 the MICs were reduced at least 2-fold for 5 of the 6 antibiotics tested, as observed when comparing the S and L strains (Fig. 5a). Ciprofloxacin and tobramycin are routinely used for the treatment of *P. aeruginosa* infections, often leading to the development of resistance to quinolones and aminoglycosides.

Together with the increased growth rates, all L strains decreased their MICs for ciprofloxacin (3.3—8.8—4.0-fold respectively strain 141_L, 10_L and 427.1_L) and strains 10 and 427.1 decreased their MIC (2.0- and 2.7-fold) for tobramycin (Fig. 5a). Screening for reversions of mutations in genes commonly associated with resistance to the tested antibiotics, which could explain the reduction in the MICs, however, revealed no changes

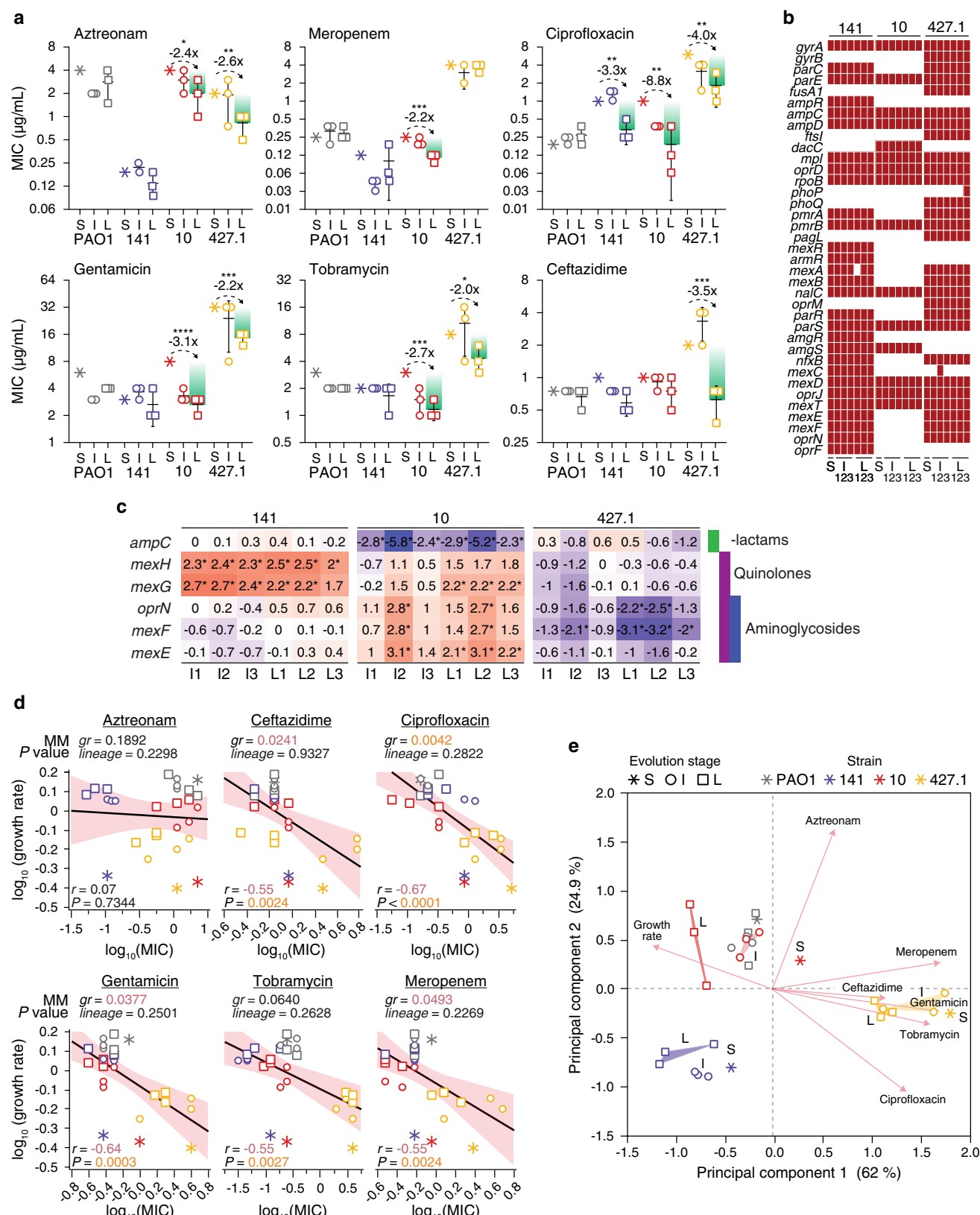

that took place during ALE at genetic level (Fig. 5b). Similarly, expression of genes known to be involved in antibiotic resistance showed only limited changes at transcriptional level (Supplementary Fig. 10). The reduction in the expression of the *ampC* gene in strains 10_I and 10_L, might explain the lower MIC for

aztreonam, while in strains 427.1_L the reduced expression of *mexEF-oprN* genes might contribute to lower the MIC for quinolone and aminoglycoside antibiotics (Fig. 5c). However, in strains 141 and 10, the expression of *mexGH* and *mexEF-oprN*, respectively, was increased relative to the S strains which is in

**Fig. 5 Increased antibiotic sensitivity during ALE in the absence of antibiotics. a** Minimum inhibitory concentrations (MIC) for the clinically relevant antibiotics aztreonam, ceftazidime, ciprofloxacin, gentamicin, tobramycin and meropenem. Differences between the starting (S) and late (L) strains were calculated by two-tailed Student $t$-test ($P$ values: *<0.0332; **<0.0021; ***<0.0002; ****<0.0001; $n = 3$ biologically independent strains) and the fold-change is reported. Only differences of >2 were considered biologically significant. Data are presented as mean values ± SD. **b** Maintenance of mutations on genes involved in antibiotic resistance during ALE. The red blocks indicate the presence of a nonsynonymous mutation (missense and nonsense SNPs and indel mutations) in the genomes of the evolved strains. **c** Normalized expression profile of genes involved in the resistance to β-lactam, quinolone and aminoglycoside antibiotics. Genes with a $\text{Log}_2(\text{FoldChange}) \geq |2|$ and an adjusted $p$-value $\leq 0.01$ (negative binomial distribution testing) in the I and L strains relative to the S strain are considered statistically different and are indicated by an asterisk. The full list of genes involved in antibiotic resistance is presented in Supplementary Fig. 10. **d** Pearson's correlation analysis of the trade-off between antibiotic resistance and growth rate. The black line and the red area represent the linear relationship between the variables and the 95% confidence interval. The Pearson's correlation coefficient ($r$) and the $P$ value are indicated at the bottom of each graph. The effect of the lineage, which depends on the specific characteristics of each starting strain, and the effect of the growth rate on the reduced MIC was evaluated by mixed model (MM) analysis. The growth rate and the MIC for each antibiotic were set as fixed variables and the lineage as random variable. The growth rate ($gr$) effect $P$ value and the lineage effect Wald $P$ value are reported at the top of each graph. In black are indicated coefficients >0.05 (not significant), in red coefficients <0.05 and in orange <0.01. **e** The relationship between lineage, antibiotics MIC and growth rate was visualized by principal component analysis (PCA) which explained 86.9% of the covariance on the first 2 components. The loadings (antibiotics and growth rate) and the strains are reported. Source data are provided as a Source Data file.

contrast with the reduction observed for the MIC values of quinolones and aminoglycoside antibiotics (Fig. 5c). Expression of the MexEF-OprN efflux pump can contribute to both aminoglycoside and quinolone resistance and the genes encoding for the efflux pump are under high selective pressure in the CF environment. However, Pearson's correlation analysis of the average expression profile of *mexEF-oprN* genes relative to ciprofloxacin, gentamicin and tobramycin MICs revealed no significant correlation between the variables ($-0.08$, $0.37$ and $0.31$, respectively; $P > 0.01$), indicating that changes in gene expression levels might not be the main cause of the reduced MICs. On the contrary, Pearson's correlation analysis of the individual growth rates relative to their MIC values showed a significant, albeit moderate, negative correlation (range $-0.55$ – $-0.67$; $P < 0.01$) for all antibiotics tested except aztreonam (Fig. 5d). To avoid overestimating the effect of the growth rate, and to account for intrinsic differences between the S strains (lineage-dependent effect), we built mixed models (MM) with growth rate and MIC for each antibiotic as fixed variables and the lineage as random variable. In none of the models, the lineage seems to have an explanatory impact as a variable (Wald $P >$ 0.05), making lineage-dependent effects very unlikely. Instead, growth rate has a strong effect on the ciprofloxacin MIC ($P = 0.0042$) and a weaker influence on ceftazidime, gentamicin and meropenem MIC ($P < 0.05$) (Fig. 5d). PCA separates the strains according to their overall antibiotic susceptibility profiles and growth rates, maintaining grouping according to their origin. However, both I and L strains move on PC1 toward increased growth rates strengthening the association between antibiotic susceptibility and growth rate (Fig. 5e).

Differences in the MIC, which in a limited manner can be caused by both epistatic interactions between non-antibiotic related mutations and changes in gene expression profiles, can be partially attributed to growth rate increases observed in the ALE-adapted strains.

## Discussion

*Pseudomonas aeruginosa* is a common cause of nosocomial infections, and as such it has become a health threat, enhanced by our inability to efficiently treat chronic infections such as those affecting patients suffering from CF[44]. CF airways become colonized by different microorganisms from early childhood, and among these *P. aeruginosa* is one of the dominant species, which has the potential to persist for the entire life span of the infected patient. The adaptive evolution undertaken by the bacteria during hundreds of thousands of generations creates broad phenotypic

and genomic changes optimizing the fitness of *P. aeruginosa* to the CF airways[7].

In a dynamic environment where presence of stresses, the immune system, and antibiotics can rapidly change, bacteria with increased mutational rate (hypermutators) are common. It has been proposed that hypermutator strains advance antibiotic resistance and increased virulence, persistence and metabolic adaptation[21]. Due to the different mutational rate caused by hypermutability, their trajectory of evolution can take distinct paths than normal strains. However, in our work, hyper- and normo-mutators show similar phenotypes indicating that they can occupy similar fitness landscapes. Even though hypermutators have intrinsically higher background genomic noise, i.e. mutations with no or limited effect, the genomic and transcriptional solutions that could be achieved to increase the cellular fitness is shared between the bacteria. Indeed, pathoadaptive mutations, which are considered the main drivers of adaptation, overlap between hyper- and normo- mutator bacteria indicating similar targets of evolution[14,23,25]. Therefore, while the results presented here may be limited to the analyzed hypermutator strains, similar trajectories of evolution, in reverse direction, might be embraced by normomutators in similar conditions making our results general for several clinical isolates of *P. aeruginosa*.

The adaptive processes occurring in the CF environment have been described as either dead-end evolution or adaptive radiation[30,45–47]. This implies that, from one side, long-term adaptation constrains the ability to colonize new niches and to reach new optimal fitness peaks outside the CF environment and, from the other, that *P. aeruginosa* in the CF environment becomes a new organism with a distinct set of properties, and it can no longer revert back to an ancestor-like state. In this perspective, the ALE experiments performed here distinguish between true dead-end evolution and adaptive radiation, documenting that even the most specialized strains after several hundred thousand generations in CF airways preserve its evolutionary potential when transferred to the chosen ALE conditions, although it is unlikely that they ever revert back to the ancestral strain as they would be outcompeted by other strains in natural environments.

Previously, forward-evolution of *P. aeruginosa* in CF airways has been studied at several levels through genome sequencing, genome-wide gene expression determined by RNA-sequencing, metabolomic profiling, and phenotyping (growth rates, antibiotic resistance, adhesion properties, virulence etc.)[7]. The picture that has emerged is one of broad diversity, indicating that many different evolutionary trajectories may result in increased fitness for

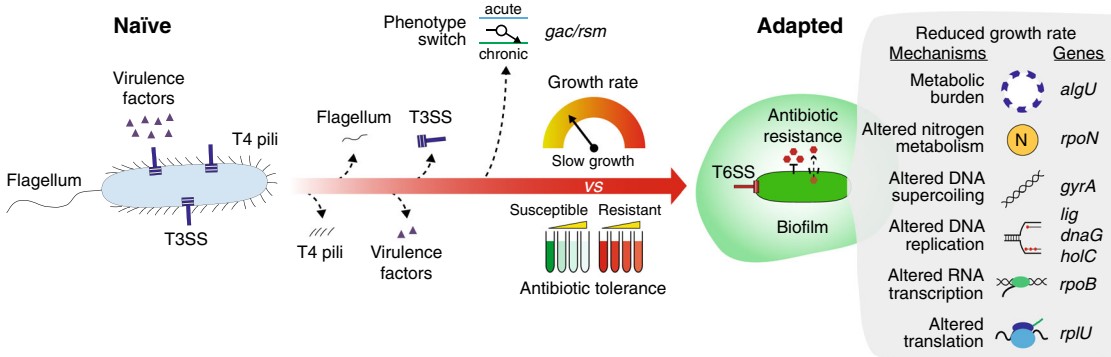

**Fig. 6 Trajectory of adaptation of *Pseudomonas aeruginosa* to the cystic fibrosis (CF) environment.** The cartoon represents the trajectories of evolution from naïve to adapted strain occurring in the CF patients. Several cellular components such as the flagellum, the type 4 pili (T4 pili), the type 3 secretion system (T3SS) and numerous virulence factors are lost during within-patient evolution due to accumulation of mutations. The Gac/Rsm regulatory system coordinates the switch between acute and chronic phenotype through changes of gene expression profiles. Meanwhile, a reduction of the growth rate occurs due to: (1) metabolic burden caused by *algU* regulator overexpression; (2) altered nitrogen metabolism caused by mutations in *rpoN* and caused by changes in the RpoN regulon transcriptional profiles; (3) altered DNA supercoiling caused by *gyrA* gene mutations; (4) altered DNA replication caused by mutations on important component of the DNA polymerase (*lig, dnaG, holC*); (5) altered transcription caused by *rpoB* gene mutations; and (6) altered translation caused by mutations in ribosomal proteins genes. Such reduction in growth rate determines an increased tolerance to antibiotics which finally helps *P. aeruginosa* to persist in the CF lungs.

the bacterial populations after colonization of the CF airways. Nevertheless, several phenotypic traits such as reduced growth rate, antibiotic resistance, increased adherence, reduced virulence, loss of motility etc. evolve in the majority of the lineages, suggesting phenotypic convergence. Moreover, both normo- and hypermutator strains show very similar phenotypes indicating parallel evolutive trajectories independently of the mutation rate[8]. Such convergence is also shown by a similar transcriptional profile in vivo independently of the contingency of evolution and the host[35]. Growth rate reduction is one of the phenotypic changes, which unexpectedly may increase persistence in the patients[5]. However, little is known about the molecular mechanisms causing it. Moreover, the trade-off between antibiotic susceptibility and growth rate/metabolic status of the cell is completely overlooked, when it comes to treating chronic *P. aeruginosa* infections[48]. Indeed, low level resistance caused by reduced growth rate may result in a sufficient fitness advantage to persist in the CF environment, before the insurgence of detectable antibiotic resistance[49]. In the case of bacteriostatic antibiotics, bacteria can become even more tolerant by reducing the number of duplications therefore promoting cell survival. In the case of bactericidal, instead, reduced growth rate reduces antibiotic uptake therefore reducing the effect of the antibiotics. However, to more broadly correlate reduced growth rate with antibiotic tolerance in *P. aeruginosa* clinical isolates, more in-deep analysis is still necessary.

A similar picture emerges from our investigation, in which a strong selective pressure for increased growth rates was established. Re-optimization of cellular physiology causing increased growth rate may, indeed, be rooted in a number of adaptive genetic alterations: (1) mutations leading to optimization of the existing metabolic networks (amino acids, sugars and oxygen) in concert with reduction of the biosynthesis of unnecessary gene products such as flagella, virulence factors, and pyoverdine; (2) mutations or changes in the transcriptional profile in genes involved in alginate production, resulting in reduction of the alginate associated metabolic burden; and (3) compensatory mutations in essential components of the cell, accumulated in a specific temporal order (TopA, RpoN, NtrC, Lig, DnaG, HolC, RpoB and RplU), which strongly alter the physiology of the cell. Thus, the limited parallel evolution observed in our work suggests

that, in reverse direction, different evolutionary trajectories may lead to decreased growth rates in the clinical strains.

However, strong signs of convergence between adaptive mechanisms associated with global regulators were also observed at both genetic and transcriptional levels. Indeed, since the targets of evolution and the selective pressures present are similar between infections, common bacterial adaptive strategies are enforced in different patients (Fig. 6)[8]. Evolving clinical strains in a new environment, different from the CF airway conditions, pushed evolution to a new fitness peak which required the reversion or compensation of several CF-specific traits allowing the identification of the most critical ones. In the CF environment, the Gac/Rsm regulatory system is pivotal for the proper switch between acute and chronic phenotype allowing survival in the CF environment and escaping the immune system[50]. The mucoid phenotype associated with AlgU regulon overexpression is often found in CF sputum sample and it is associated with higher morbidity and mortality in CF patients. Moreover, alginate might impair antibiotic penetration in biofilms being a common mechanism of adaptation to the CF[51]. Ciprofloxacin is extensively used for the treatment of CF patients and one of the leading mechanisms associated with resistance is the accumulation of mutations in the gyrase genes *gyrAB*[52]. RpoN mutations have also been identified as early mechanism of CF adaptation in *P. aeruginosa* due to their role in both nitrogen and carbon metabolism regulation[53]. Nitrogen is, indeed, a valuable resource in the CF environment and a tight regulation of its uptake is fundamental for reducing the toxicity of nitrogen oxides species which might accumulate in the CF airways[53,54]. Furthermore, several mutations in central components of the cell have also been described to impact cell physiology[40,55]. Not surprisingly, *algU* and *gyrA* are defined as pathoadaptive genes, i.e. genes in which functional mutations optimize pathogen fitness[14]. Even though AlgU, GyrA and RpoN are known targets of *P. aeruginosa* evolution in the CF environment, their role in increasing the within-patient fitness through reduction of growth rate was previously unknown[52,56,57]. Our results suggest that by tailoring global regulators functionality to the CF environment, bacteria increase their fitness in two ways: 'locally' modifying the functionality of specific system such as alginate production, nitrogen metabolism and ciprofloxacin resistance, and 'globally' by decreasing the growth rate which has

a broader impact on antibiotics tolerance and persistence. Importantly, reduction of the growth rate can occur within the first 2–3 years of the infection when evolution advances to reach higher fitness peaks through the accumulation of few regulatory mutations or changes in the expression profiles[8]. However, it is worth noting that the trajectory of evolution described in this work, from slow to fast growth, might be different from those occurred in reverse direction, from fast growth of naïve bacteria to slow growth of adapted bacteria, since several other routes of evolution are possible and evolution to a new laboratory environment does not imply reversion to a naïve-like state.

Genome sequencing of *P. aeruginosa* strains derived from infected CF patients has been introduced in CF clinics in many countries, and the absolute and precise identification of the respective clone types has already turned out to be very valuable for the clinical diagnosis and for predicting to some extent the persistence of the infection[14]. When the genome sequences, however, are additionally used to predict specific properties (phenotypes), the situation is often more complicated[58]. Many traits are indeed determined by combinations of activities of several gene products, and often these combinations have not been fully elucidated. In other cases, mutations in a gene may not result in a changed property unless epistasis occurs. Our approach combining DNA and RNA sequencing of clinical isolates and ALE to revert a clinically relevant phenotype, has allowed to identify several mechanisms of adaptation and trajectories of evolution which, even though specific for hypermutator strains, could be used by clinicians as markers of evolution and to diagnose progression of the infection towards chronicity. Whether the slow growth is selected for in the CF environment or it is simply a by-product of selection of another phenotype with higher fitness, is still less clear.

In summary, employing ALE to revert the slow growth phenotype, we were able to track compensatory mechanism which, in reverse direction, might have determined within-patient adaptation. We identified specific trajectories of evolution which converge to common processes of adaptation possibly occurring in the patient, and identified molecular mechanisms leading to high growth rate in three hypermutator strains of *P. aeruginosa*. Moreover, we showed temporal dependent steps of evolution and ultimately, the relationship between antibiotic sensitivity and growth rate. Altogether, this suggests a timeline of *P. aeruginosa* evolution in the patient (Fig. 6).

## Methods

**Bacterial strains, media and adaptive laboratory evolution**. *P. aeruginosa* clinical isolates were previously sampled and identified from sputum samples of three patient attending the Copenhagen cystic fibrosis clinic at the University Hospital, Rigshospitalet, Copenhagen, Denmark[14]. Analyses of the bacterial isolates were approved by the local ethics committee of the Capital Region of Denmark (Region Hovedstaden; registration numbers H-1-2013-032, H-4-2015-FSP). The *P. aeruginosa* laboratory strains PAO1 was used as reference. Strain *rpoN*-KO is a *rpoN* gene deletion mutant containing a gentamicin cassette in place of *rpoN* gene[56]. For ALE, each of the starting strain (141_S, 10_S, 427.1_S and PAO1) were pre-grown over night in LB medium[59] at 37 °C and the suspension used to inoculate three independent flasks containing 25 ml of LB medium. A magnetic stir bar spinning at 350 rpm was used for mixing of the culture and aeration. A total of 800 μL of culture was serially passaged using an automated liquid handling robot system after reaching stationary phase (3.2% of the total culture volume was propagated to the next culture). Four optical density measurements at 600 nm (OD$_{600}$) were taken between ODs of 0.05 and 0.95 to determine the growth rates using an in-house MATLAB package[60]. Periodically, aliquots of samples were frozen in 25% glycerol solution and stored at −80 °C for future analysis. Growth kinetics of the strains was recorded by measuring the turbidimetry at 630 nm of cultures in 96-well microtiter plates using an ELx808 Absorbance Reader (BioTek Instruments, Winooski, VT, USA) in LB medium at 37 °C. The growth rates were scored by fitting an exponential curve to the OD data recorded using the software GraphPad Prism version 8.4.3. For each strain, at least three independent biological replicates were analyzed.

**Fitness trajectories estimation**. The power-law model, used by Lenski and collaborators to explain the observations in the LTEE[33,34], was applied to the growth rate trajectories to describe the fitness increase of the populations and to predict the trajectories of evolution (Fig. 1 and Supplementary Fig. 2, 3). According to the power-law model $\mu = at^b$, the growth rate ($\mu$) is a function of the time in generations ($t$) and of the parameters $a$ and $b$. Estimation of parameters $a$ and $b$ of the models and of the 95% confidence intervals was obtained by minimizing the sum of squared errors (SSE) of the fitting by Gauss–Newton (least squares) optimization method using the statistical software JMP® version 14.3.0. Normalized root-mean-standard errors (NRMSE), used for evaluating and comparing the goodness of the fits, were calculated by dividing the RMSE of each model by the mean value of growth rate of the corresponding population. The power-law model was applied both to the combined populations (CP) for each starting strain composed by the aggregated data of the three parallel ALE experiment, and for each independent ALE experiment. To validate the accuracy of each of the models to describe and predict the trajectories of evolution, we calculated the minimum amount of growth rate data (reduced-model) necessary to describe the entire evolution trajectory without any statistically significant difference (two tailed Student's $t$ test, $P > 0.05$) in the growth rate predicted at 950 and 30,000 generations relative to the model (full-model) calculated using the entire growth rate data (Supplementary Fig. 3). The reduced-models, built using the first 200 (23% of the total; CP 141), 500 (64% of the total; CP 10) and 300 (33% of the total; CP 427.1) generations, estimated the growth rate of the populations with no difference relative to the full-model (two-tailed Student's $t$ test, $P > 0.05$; Supplementary Fig. 3) confirming that even a fraction of the growth rate data can be used to estimate fitness increases at generations not reached in our experiment.

**Comparative genomics and transcriptomics**. Genomic DNA was extracted using the DNeasy Blood & Tissue kit (Qiagen), the libraries prepared using the KAPA HyperPlus kit (Roche) and sequenced using a MiSeq machine (Illumina) with a mean coverage of 100x. Reads were trimmed, and low-quality reads and potential contamination from adapters were removed using Trimmomatic (version 0.39). Reads were mapped against *P. aeruginosa* PAO1 genome (NC_002516.2) using the BWA aligner and the MEM algorithm (version 0.7.16a) and duplicated reads marked using Picard tools "MarkDuplicates" utility (version 2.17.0). GATK (version 3.8-0-ge9d806836) was used to re-align around microindels and to call variants using "HaplotypeCaller" algorithm (setting -ploidy 1). SNPs were extracted if they met the following criteria: a quality score of at least 50, a root-mean-square (RMS) mapping quality of at least 25 and a minimum of three reads covering the position. Microindels were filtered based on a quality score of at least 500, an RMS mapping quality of at least 25 and support from at least one-fifth of the covering reads[14,35]. The genomes of the starting strains were assembled to calculate the estimated genome length using Unicycler assembler (version 0.4.8)[61]. The substitution rate for each evolved strain was calculated as bp per estimated genome length of the starting strain per replication. Phylogenetic relatedness between the evolved strains was inferred by Maximum Parsimony (MP) or Maximum Likelihood (ML) analysis using a nucleotide sequence consisting of 8065 concatenated SNP mutations for each strain. The bootstrap consensus tree was built from 500 replicates using Mega 7.0.26[62]. For whole genome RNA sequencing, representative initial and late clones of three independent ALE replicates were grown in LB medium at 37 °C in full aeration in duplicates. Cells were harvested during mid-exponential (OD$_{600}$ = 0.6) and transcription was blocked adding 1/9 volume of cold stop solution (5% H$_2$O-saturated phenol in ethanol) to 1 volume of bacterial culture in a pre-chilled collection tube. Total RNA was extracted using Trizol reagent (Thermo Fisher Scientific) and total RNA (>200 nt) was purified from the aqueous phase using the RNA Clean & Concentrator-25 kit (Zymo Research) following manufacturers' indications[35]. RNA quality and integrity were assayed using RNA Nano kit on an Agilent Bioanalyzer 2100 machine. A total of 1 μg of RNA from samples with a RIN higher than 9 were used for preparing strand-specific sequencing libraries using the KAPA RNA HyperPrep Kit (Roche) after rRNA depletion obtained using *P. aeruginosa* riboPOOL kit (siTOOLs Biotech). Sequencing was performed on an Illumina NextSeq 500 machine generating ca. 30 million reads per sample of 2 × 75 bp reads. Reads were trimmed using the Trimmomatic (version 0.39), residual rRNA reads were discarded using Sort-MeRNA (version 2.1) and mapped using BWA MEM (version 0.7.16a) to *P. aeruginosa* PAO1 genome (NC_002516.2). Mapped reads were counted using htseq-count version 0.11.2. Reads normalization was performed using the "vst()" function from the R package DESeq2 (version 1.28.1) with option "blind" set as "True". Normalized counts were used to evaluate whole transcriptome similarities between sampling using hierarchical clustering analysis (HCA), principal component analysis (PCA) and k-means clustering on PCA-reduced data. HCA was performed in R (version 4.0.4) using Spearman correlation as distance in "dist()" function and "complete" as method in hclust() function. Sample correlation and clustering were visualized as dendrograms and heat maps using the R package ComplexHeatmap (version 2.4.2). Principal component analysis on normalized reads counts was performed using prcomp() function with scale option set as "FALSE". PCA-reduced data was clustered using k-means algorithm identifying the optimal number of clusters using the NBClust function contained in the NBClust package (version 3.0) using all available indexes. Differential gene expression analysis was

performed using the R package DESeq2 (gene expression modelled using the negative binomial distribution) considering statistically significant genes with a $Log_2(FoldChange) \geq |2|$ and an adjusted $p$-value $\leq 0.01$. Regulon enrichment analysis was performed on statistically significant differentially expressed genes using the provided "term_enrich.py" script and using gene-regulon associations obtained from literature[63]. Classes with a $p$-value (Hyper-geometric test) and adjusted $p$-value (Bonferroni correction for multiple tests) $\leq 0.05$ were considered statistically significant.

**Complementation assay**. Complementation experiments of *gyrA* and *rpoN* mutations and overexpression of *algU* regulator were carried out using the pHERD30T plasmid which contains the arabinose inducible $P_{BAD}$ promoter and the *aacC1* gentamicin resistance gene[64]. The coding sequence of the *algU* gene deriving from strain 141_S was amplified using primers algU_usr_(pHERD30T)_F and algU_usr_(pHERD30T)_R and cloned by USER technology into the pHERD30T plasmid amplified using primers pHERD30T_usr_(algU)_F and pHERD30T_usr_(algU)_R (Supplementary Table 1). The coding sequence of the *gyrA* gene deriving from strains 10_S and PAO1 was amplified using primers gyrA_usr_(pHERD30T)_F and gyrA_usr_(pHERD30T)_R and cloned by USER technology into the pHERD30T plasmid amplified using primers pHERD30T_usr_(gyrA)_pHERD30T_usr_(gyrA)_R (Supplementary Table 1). The coding sequences of the *rpoN* gene deriving from strains 10_S, 10_L1, 10_L3 and PAO1 were amplified using primers rpoN_usr_(pHERD30T)_F and rpoN_usr_(pHERD30T)_R and cloned by USER cloning technology into the pHERD30T plasmid amplified using primers pHERD30T_usr_(rpoN)_F and pHERD30T_usr_(rpoN)_R (Supplementary Table 1). All PCRs were performed with Phusion U Hot Start DNA Polymerase (Thermo Fisher Scientific Inc., Waltham, USA) and USER cloning performed with USER™ Enzyme (New England Biolabs Inc., Ipswich, USA) accordingly to the manufacturer protocol. The resulting pHERD30T plasmids were transferred to *P. aeruginosa* strains by either chemical transformation or tripartite conjugation using the pRK600 plasmid as helper[65]. For complementation of *gyrA* and *rpoN* genes mutations, either 0% or 0.1% of arabinose was added to the cell culture to induce the expression of the gene.

**Motility assay**. Motility in liquid cultures was assayed using the sedimentation assays[66]. Bacteria were inoculated in 3 ml of King's Medium B and cultured at 37 °C at 250 rpm. After 48 h the $OD_{600}$ of the cultures was measured (ref_OD), and this value was set to 100%. The tubes were then maintained without shaking to allow sedimentation of the strains as function of their motility. After 24 h a sample was taken from the top part of the culture to measure the $OD_{600}$ and to estimate the sedimentation relative to the ref_OD. For each strain, three independent biological replicates were analyzed.

**Biofilm assay**. Biofilm production was assayed by measuring attachment to NUNC peg lids[8]. Strains were grown in 96-well plates with 150 μl of LB medium at 37 °C at 150 rpm. Peg lids (NUNC cat no. 445497) were used instead of the standard plate lids. After 24 h the $OD_{600}$ was measured and used as reference of planktonic growth. The peg lids were washed with 180 μl PBS and transferred for 15 min to a microtiter plate containing 160 μl of 0.01% crystal violet. The peg lids were then washed three times with 180 μl PBS to remove unbound crystal violet. The peg lids were transferred to a microtiter plate containing 180 μl of 99% ethanol, to detach the adhering cells from the peg lid and to read the $OD_{590}$. For each strain, three independent biological replicates were analyzed.

**Pyoverdine assay**. Pyoverdine production was assayed by growing bacteria in King's B medium[67]. Bacteria were cultured in 3 mL King's B medium for 48 h at 37 °C. After incubation, supernatant samples were collected, and the fluorescence measured at 400/460 nm excitation/emission on a Synergy H1 Hybrid Multi-Mode Reader (BioTek Instruments, Winooski, VT, USA). Fluorescence readings were normalized by the absorbance of the cell culture ($OD_{600}$). For each strain, three independent biological replicates were analyzed.

**Antibiotic susceptibility test**. The Minimum Inhibitory Concentrations (MICs) for aztreonam, ceftazidime, ciprofloxacin, gentamicin, tobramycin and meropenem were measured by E-tests according to the European Committee on Antimicrobial Susceptibility Testing (EUCAST) guidelines. For Pearson correlation and Principal Component (PCA) analysis, MIC and growth rate data were normalized by log10 transformation to reduce the lineage specific differences. Pearson correlation was analyzed by Row-wise method and PCA was analyzed on covariances. Mixed models of the relationship between growth rate, MIC and lineage were computed using MIC as output, growth rate as fixed effect and the lineage (strains PAO1, 141, 10 and 427.1) as random categorical variable. The Wald $P$ value was computed on the variance of the random variable effect, while $P$ value were computed on the growth rate variable effect. Analyses were carried out using the statistical software JMP® version 14.3.0.

**Reporting summary**. Further information on research design is available in the Nature Research Reporting Summary linked to this article.

## Data availability

Sequencing data that support the findings of this study have been deposited in the EMBL-EBI European Nucleotide Archive (ENA) with the primary accession code PRJEB38310. Source data are provided with this paper. Virulence factor genes were obtained from the Victors (http://www.phidias.us/victors/)[68], VFDB (http://www.mgc.ac.cn/VFs/)[69] and PseudoCAP (https://www.pseudomonas.com/pseudocap)[70] databases. Source data are provided with this paper.

## Code availability

A thoroughly commented analysis of the RNA-seq and mutational data, and all necessary code and resources to replicate the results are available in the Zenodo repository with the identifier DOI: 10.5281/zenodo.3612820 (https://doi.org/10.5281/zenodo.3612820).

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

## Acknowledgements

This research was funded by the "Cystic Fibrosis Foundation" (CFF), grant number MOLIN18G0, the "Cystic Fibrosis Trust", grant number Strategic Research Centre Award—2019—SRC 017, by the "Novo Nordisk Foundation Center for Biosustainability (CfB)", Technical University of Denmark and by "The Novo Nordisk Foundation", NNF grant number NNF10CC1016517. H.K.J. was supported by The Novo Nordisk Foundation as a clinical research stipend (NNF12OC1015920), by Rigshospitalets Rammebevilling 2015–17 (R88-A3537), by Lundbeckfonden (R167-2013-15229), by Novo Nordisk Foundation (NNF15OC0017444), by RegionH Rammebevilling (R144-A5287),

by Independent Research Fund Denmark/Medical and Health Sciences (FTP-4183-00051) and by 'Savværksejer Jeppe Juhl og Hustru Ovita Juhls mindelegat'.

## Author contributions

R.L.R. and S.M. designed the study. R.L.R. and E.R. carried out the experimental work, analyzed the data and prepared the figures. A.M.F. supervised and designed the adaptive laboratory evolution aspects. H.K.J. provided the clinical strains. R.L.R. wrote the manuscript with input from all authors.

## Competing interests

The authors declare no competing interests.
