## [Peer Review File · Nature Communications]

REVIEWER COMMENTS

Reviewer #1 (Remarks to the Author):

The manuscript "Reverse evolution of slow growth phenotype reveals evolutionary trajectories to chronic infection in *Pseudomonas aeruginosa*" presents an interesting approach to the study of clinical mutations and their role in the fitness of strain. The potential for information regarding the evolution of clinical isolates in chronic infection is significant. While the paper does identify specific genes that appear to influence bacterial growth rate, further experimentation and validation needs to be performed in order to draw any conclusions with regard to in patient adaptation. In general, the significance of the findings are over stated and the underlying assumption that ALE in LB will result in "adaptive trajectories back to the naïve-like state of the ancestor" is unsubstantiated. While the analysis appears to be sound and appropriate, the following major concerns remain:

1. The concept of reverse evolution is a paradox, in that the work done in this manuscript rather demonstrates adaptive evolution to a laboratory growth environment. Furthermore, as these clinical strains most likely did not arise from a laboratory environment one cannot state that the growth conditions reflect the conditions of the ancestral strain. Lastly, without a genetically constructed ancestral strain one cannot comment on the reversion of the isolates to their ancestral state. PAO1 is effectively a laboratory strain and was not originally isolated from the environment of an early CF infection.
2. The use of hypermutator strains primed this study for the identification of mutations as in any condition hypermutator strains will generate mutations which will be selected based on a competitive fitness advantage.
3. The work presented in this paper could be enriched by providing context to the extent in which similar mutations have been identified in other *Pa* patient populations.
4. Growth phenotypes and their role in antibiotic efficacy is an established phenomenon. Furthermore, at certain times the authors confuse the terms tolerance and resistance. Behaviors described in this paper relating to growth rate point to tolerant bacteria because they can persist in the presence of antibiotics but cannot grow nor do they have a genetic mutation that renders the antibiotic target ineffective.
5. Genes such as *algU* and *mucA* along with the loss of T3SS, T4 pili, and flagellum have already been established as having a significant role during chronic host infection but not during laboratory growth conditions.
6. The evolution experiments lacked controls such as a non-hypermutator clinical isolate and non-clinical hypermutator strain. Results from these controls would validate the results observed in the experimental isolates.
7. Complementation experiments lacked controls including expression of genes in PAO1 to see if the same effect would be observed in non-evolved strains. Since the genetics of the evolved clinical isolates are very complicated placing mutations in a WT background provides a context to the significance of the mutation in isolation.

Reviewer #2 (Remarks to the Author):

In "Reverse evolution of slow growth phenotype reveals evolutionary trajectories to chronic infection in *Pseudomonas aeruginosa*" La Rosa et al., present a very thorough and systematic study with several important insights on the different adaptive strategies that PA may take to adapt to the CF lung. The study combines experimental evolution with genomics, transcriptomics, functional genomics and MIC testing to elucidate whether strong selection to increase growth rate in vitro can revert phenotypically, genetically, or both, the slow growth typically associated with adaptation to a CF environment.

From their systematic approach the authors generally find that:

- Clinical isolates maintain their evolvability, yet there are strain specific life-history traits that influence the rate at which fitness can increase. This is the result of the strains following different mutational trajectories during their adaptation to the patient.

- The different strains had to revert distinct metabolic or regulatory systems to gain fitness levels similar to those of the laboratory reference PAO1. In one case the burden of alginate production was alleviated by an epistatic effect reducing the overexpression of algU. In another case the cumulative effects of mutations in topA and rpoN might have restored growth rate in one of the clinical isolates. In the isolate having the lowest fitness improvement, the accumulation of multiple potential compensatory mutations seems to have restored growth rate, at least partially. In the last case, the mechanism is not as clear as for the other.

- However unique the adaptive path was at the genomic level, there were also insights of convergent evolution across the different isolates, potentially by transcriptional changes in global regulators that influence multiple process at once.
- The increase in growth rate was associated with a decrease in MIC against 5 of 6 commonly used antibiotics in patients suffering of CF.

These are relevant observations from a very complete and systematic study approach. The manuscript is very clear and easy to read, and I particularly appreciate the structure of the results, making it easy to follow the different insights gained from the experiments. I believe this is a great manuscript adding some important insights as to how PA adapts to CF environment that would nicely complement the literature on this area. I only have very minor comments:

1. Can the authors explain more rigorously why the experiments were done with hypermutators? In the manuscript it is said that this was chosen to "enhance evolvability and select for the highest fitness" lines 77-78. I think the general audience would benefit from a more complete explanation of this choice. Also, I think it would also be important to discuss how the results might differ if the clinical isolates were not hypermutators. In my opinion this, together with the fact that there were only three clinical isolates and three evolved populations per time point, also means that the authors could phrase their conclusions a bit less generally as it is limited to a very specific (albeit very important) group of strains affecting CF patients.

2. The 427 clinical isolate is particularly interesting because it has the lowest fitness improvement, the least straightforward genomic mechanism of reversal and the highest antibiotic resistance (at least the S strain). Can the authors say more about the underlying genomic differences among the S strains and how they inferred that the mutations mentioned in lines 257-270 are potentially compensatory?

3. The decrease in MIC despite retaining the mutations in known resistance genes against certain drugs is puzzling (i.e. gyrAB and nfxB conferring resistance against CIP). Given the vast amount of mutations accumulated it is certainly a challenge to determine whether this is an epistatic effect or if there is some additional compensatory process. Perhaps the authors can discuss this a bit further, or perhaps they can identify something at the transcriptional level that could account for this? as it stands right now, there is no discussion about this.

4. In a related point to the previous one, I wonder if the authors can relate some of the MIC and growth rates observed in the evolved populations with their estimated duration of cumulative exposure to the different antibiotics within the patients? Or if the duration of adaptation within the single patient may also play a role into the potential of the strains to be resistant to multiple drugs and to improve their growth rate.

Very minor point:

Fig 4 in my opinion is a bit inconsistent and redundant. For some cases there are counts and trees, for others only trees which are presented somehow differently too. This is not a big issue, but I wonder if the authors can make the figure more consistent or at least less loaded.

Reviewer #3 (Remarks to the Author):

This manuscript describes a laboratory evolution experiment that studies compensatory evolution in *Pseudomonas aeruginosa*. Founding strains with a slow growth phenotype were propagated in a rich environment to allow for adaptive evolution and increased growth rate. The genomes of evolved strains were sequenced to identify accumulated mutations, and differential gene expression patterns were also determined. Different founding genotypes took different adaptive

trajectories during compensatory evolution. However, some of the similarities suggested that some global regulators were involved in the original slow growth phenotype, and the effects of some specific genes were confirmed by genetic analyses. In addition, the increased growth rate was correlated with a restoration of antibiotic sensitivities, suggesting a direct trade-off. Most of the methods and results are described sufficiently. Overall, the manuscript is well written and the figures nicely summarize the data and the main points. My main concerns are discussed below, along with some more specific comments on a line-by-line basis.

General concerns:

How common is reduced growth rate in CF-adapted strains? It may be a phenotypic marker of adaptation, but how important is it when compared to other phenotypic markers like mucoidy, etc?

More information should be given on the overall phenotypes of the founding strains and the evolved strains. For example, did phenotypes such as mucoidy, motility, biofilm formation, or pigment production change during the experiments? These are important to consider because propagation in broth alone may select for certain phenotypes, which can result in gene expression changes. For example, growth in broth selects for reduced mucoidy, so changes in alginate gene expression patterns may have arisen to that form of selection, not selection on increased growth rate per se. Similarly, growth in broth selects for reduced motility and biofilm formation, so flagella and pellicle gene expression may have been modified. So the question is: Did the laboratory evolution select for increased growth rate, or was it a by-product of selection for other factors?

Why were hypermutator strains chosen? I understand that the increased rate of mutation supply can speed up the adaptive process, but many laboratory evolution experiments have been successfully performed with standard non-mutators. The problem here is that mutators can take adaptive trajectories that non-mutators may not. Not only is the mutation rate an order of magnitude greater, but mutators also have a different spectrum and directional bias of mutations. Furthermore, the greatly increased mutation rate adds noise to the mutation patterns, so some of the observed mutations may be insignificant and play no future role in adaptation. How these differences between mutators and non-mutators could impact the interpretation of the results should be discussed.

The results of genome sequencing and differential gene expression identified thousands of potential genes of interest. The fact that these were mutator strains definitely added to the wealth of candidate genes. However, the method used to select genes for further characterization is not clearly described. It seems that the authors focused primarily on genes or pathways that were already known to be important for *P. aeruginosa* adaptation to CF patients. The authors do state that "only some of the genetic changes correlating with an increase in growth rate will be presented" (line 188-189), but it would be useful to know how representative these discussed changes are of the total genetic changes. In other words, were these pathways objectively the most important, or were they chosen subjectively from a large number of options?

A discussion of bacteriostatic versus bacteriocidal antibiotics should be presented here. Bacteriostatic antibiotics only affect cells that are actively dividing, so reduced cell division can promote cell survival. This is one explanation of how reduced growth rate can cause antibiotic resistance.

In multiple cases (see below), comparisons are made between evolved 10, 141, and 147.1 strains and the PAO1 reference genome, but this is not appropriate and can add confusion to the interpretation of the results. Only the evolved PAO1 strains should use the PAO1 reference genome as the ancestor reference. For evolved 10, 141, and 147.1 strains, the 10S, 141_S, and 147.1_S genomes should be used as the ancestor reference, respectively.

Specific comments (line number):

102: Is growth rate in "cell doublings per hour"?

105: The graph in Fig 1a shows that the S version of PAO1 had a slower growth rate than the I or

L versions.

129-132: This concept is known as the contingency of evolution.

139-140: Each founder S strain is different from each other, and different from PAO1, so the number of mutations between an S strain and PAO1 is not relevant at this point. I would remove this comparison, and remove these data from Figure 2a to avoid confusion. Also, I would call these differences "SNPs", not "mutations". The important comparison is between the founder S strain and the evolved I and L strains.

154-156: I have multiple points for tree in Figure 2c.

1) Why was maximum parsimony used, rather than maximum likelihood?

2) The phylogenetic tree in Figure 2c assumes the terminal taxa are contemporaries, but that is not the case because S evolved into I, and I into L. So can this be displayed in a different way, one that reflects these relationships?

3) If a tree is still used, then a separate tree should be constructed for each strain group, with the S version as the root. Again, the relationships between the founding strains (and PAO1) are not relevant and will only add confusion to the interpretation.

4) Why are the expected relationships within strain groups not recovered? For example, if each replicate in each strain group is evolving independently, I1 should be more closely related to L1, and I2 to L2, and I3 to L3, correct? That does not appear to be the case. Also, since number of generations of evolution should be correlated with number of accumulated mutations, all I (intermediate) isolates should have less mutations than L isolates, but that does not appear to be the case either.

167-169: Based on Figure 2d, only Strain 10 evolved towards the area occupied by PAO1. The other 2 strains are distinct in PCA-space, so why was Strain 141 said to evolve towards PAO1 as well?

201-204: Were there any mutations within the *algU* and *mucA* genes themselves?

252-255: Much like the *topA* mutations, the effects of the *ntrC* mutations are highly speculative and could use some more genetic support.

265: Recent studies with *P. aeruginosa* have found that ribosomes and DNA replication machinery are under selection in isolates from CF patients.

317: In Figure 4g, I assume the asterisks indicate SNPs between S strains and reference PAO1. Again, this comparison should be removed because PAO1 is not the ancestor of the S strains, and this can confuse the interpretation of the comparison between S and ALE.

333: On average, the intrinsic sensitivity of PAO1 and 141 is not much higher than for strain 10. The trend is really being driven by the high resistance strain 427.1.

337-340: Interesting to find a lack of mutations in well-know resistance genes, suggesting the compensatory mutations were located in other unknown genes.

407-408: I would be careful with this statement. Just because the increase in growth rate took different trajectories, it doesn't mean the original decrease in growth rate took different trajectories. In theory, the same genotype can follow different adaptive strategies based on mutation supply, and the likelihood of this would increase in these mutator strains.

452-454: This may be an over-statement of the confidence in the conclusion. These experiments can suggest which mechanisms caused the slow growth rate, but they cannot determine if the slow growth rate was the target of selection, or if it is a by-product of selection on another phenotype.

518: Gene expression patterns were determined for growth in aerated LB broth. Studies have demonstrated that gene expression patterns in a medium that more closely simulate the CF lung environment may be quite different from that in rich LB medium. It is possible that the slow growth phenotype interacts specifically with characteristics of the CF lung environment, and these may not be captured in LB. The authors may want to discuss this as a potential limitation.

546-551: This paragraph highlights one of the issues with using hypermutable strains for adaptive evolution experiments. The greatly increased mutation rate adds so much noise to the mutation patterns, so that the importance of the observed mutations cannot be determined. Newly generated mutations may not have been exposed to selection yet, so they can be observed even though they will not play a role in adaptation. To compensate, the authors end up filtering out and ignoring a large proportion of the mutations. So although some evolved 141 strains have over 2800 mutations, they focused on 17 genes from these evolved lines.

924: Figure S3 panels b,c,d - Again, the comparison of strains back to reference PAO1 should be removed. For panel a, this comparison makes sense because PAO1 is the ancestor of the evolved strains, and there are no SNPs at generation 0. For panels b,c,d, the reference ancestor should be the respective S strain, and there should also be no SNPs generation 0.

REVIEWER COMMENTS

Reviewer #1 (Remarks to the Author):

The manuscript “Reverse evolution of slow growth phenotype reveals evolutionary trajectories to chronic infection in *Pseudomonas aeruginosa*” presents an interesting approach to the study of clinical mutations and their role in the fitness of strain. The potential for information regarding the evolution of clinical isolates in chronic infection is significant. While the paper does identify specific genes that appear to influence bacterial growth rate, further experimentation and validation needs to be performed in order to draw any conclusions with regard to in patient adaptation. In general, the significance of the findings are over stated and the underlying assumption that ALE in LB will result in “adaptive trajectories back to the naïve-like state of the ancestor” is unsubstantiated. While the analysis appears to be sound and appropriate, the following major concerns remain:

1. The concept of reverse evolution is a paradox, in that the work done in this manuscript rather demonstrates adaptive evolution to a laboratory growth environment. Furthermore, as these clinical strains most likely did not arise from a laboratory environment one cannot state that the growth conditions reflect the conditions of the ancestral strain. Lastly, without a genetically constructed ancestral strain one cannot comment on the reversion of the isolates to their ancestral state. PAO1 is effectively a laboratory strain and was not originally isolated from the environment of an early CF infection.

We agree with the reviewer that the ALEs presented in this manuscript demonstrate adaptation to a laboratory environment. However, we defined our approach as reverse evolution since the goal of our work is to revert the slow growth phenotype of clinical isolates independently of the ancestral strain condition. We have changed the sentences reporting “adaptive trajectories back to the naïve-like state of the ancestor” in “adaptive trajectories back to the high growth rate of naïve bacteria”. Even though the growth condition of the ALE doesn’t reflect the condition of the ancestor strains, the growth rate of clinical strain isolated from the same patients show a growth rate in lab condition comparable to the one of PAO1, indicating that the growth rate reached in our ALE is aligned with the growth rate of the ancestor strains. We have included this information in the text (lines 130-133).

Due to the complexity of the genetics of the clinical strains, it has not been trivial to modify these bacteria. In literature there are very few publications where clinical strains are genetically modified, since several genetic engineering tools commonly used for laboratory strains are not working. We tried to revert the mutations in the evolved strains using both recombineering (Wirth et al., 2019 doi: 10.1111/1751-7915.13396) and two-step allelic exchange (Hmelo et al., 2015 doi: 10.1038/nprot.2015.115) without success. For this reason, we switched to complementation assay which allowed us to confirm the role of the identified mutations in the increase in growth rate of the evolved strains.

2. The use of hypermutator strains primed this study for the identification of mutations as in any condition hypermutator strains will generate mutations which will be selected based on a competitive fitness advantage.

We decided to use hypermutator strains for three different reasons: 1) due to the increased mutation supply, the ALE experiments would have been shorter than with normomutator strains; 2) genes under high selective pressure in the CF environment are equally mutated in normomutator and hypermutator bacteria and indeed pathoadaptive genes are shared between the groups; 3) hypermutator strains reach similar fitness peaks of normal mutator strains showing similar phenotypes. Now this is clearly discussed in different parts of the text. In addition, we performed phenotype assays to assess whether our conditions were selecting for increased growth rate or other unrelated phenotypes (lines 156-163).

3. The work presented in this paper could be enriched by providing context to the extent in which similar mutations have been identified in other *Pa* patient populations.

We have clarified the relevance of similar mutations, by adding a new supplementary figure (Fig. S10) showing the distribution of the identified mutations in our collection of 474 clinical strains of *P. aeruginosa* and by adding a relevant reference (Klockgether et al., 2017 doi: 10.1165/rcmb.2017-0356OC). Importantly, mutations in *algU*, *gyrA* and *rpoN* have been previously identified in many clinical isolates of *P. aeruginosa*, however, their role in the decrease in growth rate was previously not assessed.

4. Growth phenotypes and their role in antibiotic efficacy is an established phenomenon. Furthermore, at certain times the authors confuse the terms tolerance and resistance. Behaviors described in this paper relating to growth rate point to tolerant bacteria because they can persist in the presence of antibiotics but cannot grow nor do they have a genetic mutation that renders the antibiotic target ineffective.

We have modified the paragraph related to the antibiotic resistance to account for the reviewer's comment (lines 396-442) and extended the discussion to underline the role of low level resistance in CF (lines 496-503).

5. Genes such as *algU* and *mucA* along with the loss of T3SS, T4 pili, and flagellum have already been established as having a significant role during chronic host infection but not during laboratory growth conditions.

As indicated by the reviewer, the identified genes have already been described to be important in the CF environment. However, their role in causing a slow growth phenotype was previously unknown. Using our approach we were able to define the connection between growth rate and genes such as *algU*, *rpoN* and *gyrA* and to track regulatory changes influencing the lifestyle of *P. aeruginosa* both in laboratory condition and in the patient.

6. The evolution experiments lacked controls such as a non-hypermutator clinical isolate and non-clinical hypermutator strain. Results from these controls would validate the results observed in the experimental isolates.

We believe that even though the trajectories of evolution of hyper and normal mutator strains might be different, similar targets of mutations (*algU*, *topA*, *gyrA* and

rpoN) have been identified as being under high selective pressure in the CF environment in both normo- and hyper- mutator strains. This indicates that even though our study might be limited by the specific strains used and for hypermutator strains, it recapitulates events occurring in many CF infections and in many different lineages of *P. aeruginosa*. Moreover, in the case of strains 141, 10 and 427.1, all the strains in our collection isolated from the same patients have all the same mutations in *mutS* and *mutL* genes, therefore being all hypermutators. This makes it impossible to carry out the experiment with non-hypermutator clinical strains. We have included a new figure showing both the percentage of patients having the first isolate hypermutator and the percentage of clone types having a first strain hypermutator in our collection from which strains 141, 10 and 427.1 were selected (Fig. S10a; lines 112-117).

Recently, Grekov and colleagues (Grekov et al., 2020 doi: 10.1038/s41396-020-00841-6) performed a similar experiment evolving in LB medium a hypermutator laboratory strain of *P. aeruginosa*, observing similar mutations and changes in gene expression profiles, therefore, validating the results obtained with our clinical isolates.

7. Complementation experiments lacked controls including expression of genes in PAO1 to see if the same effect would be observed in non-evolved strains. Since the genetics of the evolved clinical isolates are very complicated placing mutations in a WT background provides a context to the significance of the mutation in isolation.

As suggested by the reviewer, we have performed the complementation experiments in PAO1 (Fig. S7). In the case of *algU* overexpression, the result was comparable to the 141_I1 and 141_L1 strains. In the case of *gyrA* and *rpoN* genes, instead, complementation was, as expected, not achieved since the context and the presence of the mutant genes is required for the complementation. A dominant effect of the wild type copies was, therefore, observed. In support of our results with the evolved clinical strains, we have included the growth rate of a *rpoN* deletion mutant which showed a similar reduction of the growth rate as the clinical strain 10_S (Fig. S7f).

In consideration that the complementation of *rpoN* in strain 10_S was not achieved due to the lack of a properly functional gyrase/topoisomerase system (*gyrA/topA* mutations), we think that the mutations identified can only be resolved in the context of the complex background of the clinical strains, being epistatic interactions fundamental for the complementation of the phenotype. In our group, other mutations in *gyrA* gene were shown not to be directly linked to reduced growth rate in the clean background of PAO1, even though a strong correlation was shown in clinical isolates. Moreover, an attempt to correlate *gyrA* mutations to biofilm formation in the clean background of PAO1 failed, due to the lack of epistatic interactions which are only present in the complex background of the clinical isolates (Bartell et al., 2019 doi: 10.1038/s41467-019-08504-7). Furthermore, even though we presented in the manuscript the results of the complementation assays in presence of 0.1% arabinose, a similar result was obtained in absence of inducer. This further confirms that even very low levels of both GyrA and RpoN proteins, available in the cell due to the intrinsic leakage of the expression system used, are enough to re-establish the growth rate phenotype.

We have included these results in a new supplementary figure (lines 244, 289, 301-303, 307-309; Fig. S7).

Reviewer #2 (Remarks to the Author):

In “Reverse evolution of slow growth phenotype reveals evolutionary trajectories to chronic infection in *Pseudomonas aeruginosa*” La Rosa et al., present a very thorough and systematic study with several important insights on the different adaptive strategies that PA may take to adapt to the CF lung. The study combines experimental evolution with genomics, transcriptomics, functional genomics and MIC testing to elucidate whether strong selection to increase growth rate in vitro can revert phenotypically, genetically, or both, the slow growth typically associated with adaptation to a CF environment.

From their systematic approach the authors generally find that:

- Clinical isolates maintain their evolvability, yet there are strain specific life-history traits that influence the rate at which fitness can increase. This is the result of the strains following different mutational trajectories during their adaptation to the patient.
 - The different strains had to revert distinct metabolic or regulatory systems to gain fitness levels similar to those of the laboratory reference PAO1. In one case the burden of alginate production was alleviated by an epistatic effect reducing the overexpression of algU. In another case the cumulative effects of mutations in topA and rpoN might have restored growth rate in one of the clinical isolates. In the isolate having the lowest fitness improvement, the accumulation of multiple potential compensatory mutations seems to have restored growth rate, at least partially. In the last case, the mechanism is not as clear as for the other.
- However unique the adaptive path was at the genomic level, there were also insights of convergent evolution across the different isolates, potentially by transcriptional changes in global regulators that influence multiple process at once.
- The increase in growth rate was associated with a decrease in MIC against 5 of 6 commonly used antibiotics in patients suffering of CF.

These are relevant observations from a very complete and systematic study approach. The manuscript is very clear and easy to read, and I particularly appreciate the structure of the results, making it easy to follow the different insights gained from the experiments. I believe this is a great manuscript adding some important insights as to how PA adapts to CF environment that would nicely complement the literature on this area. I only have very minor comments:

1. Can the authors explain more rigorously why the experiments were done with hypermutators? In the manuscript it is said that this was chosen to “enhance evolvability and select for the highest fitness” lines 77-78. I think the general audience would benefit from a more complete explanation of this choice. Also, I think it would also be important to discuss how the results might differ if the clinical isolates were not hypermutators. In my opinion this, together with the fact that there were only three clinical isolates and three evolved populations per time point, also means that the authors could phrase their conclusions a bit less generally as it is limited to a very specific (albeit very important) group of strains affecting CF patients.

We appreciate the reviewer's comment and we have, therefore, expanded the introduction with information about the distribution and role of hypermutator strains in CF infections (lines 68-76). Similarly, we have discussed how hypermutator strains differ from normal mutator strains in view of the results obtained in our research (lines 454-468).

2. The 427 clinical isolate is particularly interesting because it has the lowest fitness improvement, the least straightforward genomic mechanism of reversal and the highest antibiotic resistance (at least the S strain). Can the authors say more about the underlying genomic differences among the S strains and how they inferred that the mutations mentioned in lines 257-270 are potentially compensatory?

We agree with the reviewer that strain 427.1 shows the most interesting profile. The strain underwent a long evolutionary history of more than 30 years in patients, therefore, the genetic context is much more complex than the other strains. Candidate genes were selected according to the criteria presented in materials and methods. Specifically for strains 427.1, after performing enrichment analysis, we looked at genes with a fundamental role in the cell physiology (lines 317-323). We moved the criteria for selection of the mutations from Materials and Methods to the main text to be more accessible to the reader (lines 224-233). There are substantial genomic differences between the S strains. Indeed, the strains belong to 3 distinct clone types (more than 10,000 SNPs of difference) which were previously sequenced and the phenotype analyzed in our group (Marvig et al., 2015 doi:10.1038/ng.3148; Bartell et al., 2019 doi:10.1038/s41467-019-08504-7). We have included more information about the starting strains in the result section (lines 119-122).

3. The decrease in MIC despite retaining the mutations in known resistance genes against certain drugs is puzzling (i.e. *gyrAB* and *nfxB* conferring resistance against CIP). Given the vast amount of mutations accumulated it is certainly a challenge to determine whether this is an epistatic effect or if there is some additional compensatory process. Perhaps the authors can discuss this a bit further, or perhaps they can identify something at the transcriptional level that could account for this? as it stands right now, there is no discussion about this.

As suggested by the reviewer, we looked at changes in the transcriptional profile of genes causing resistance to the classes of antibiotics used in our study to identify other possible mechanisms of reduced MIC. We were able to identify some possible explanations as for example the reduction of aztreonam MIC in strains 10 as consequence of reduced *ampC* gene expression, and the reduced MIC toward ciprofloxacin, tobramycin and gentamicin in strains 427.1 due to reduced expression of the *mexEF-oprN* multidrug efflux pump. However, we did not find a general and convergent mechanism for the reduction in the MIC. Indeed, the same *mexEF-oprN* pump was overexpressed in I and L strains of 10 and 141 indicating that MexEF-OprN might not be the cause of the reduced MIC. We have included these results and a new supplementary figure (Fig. S9) and in the text (lines 397-442).

4. In a related point to the previous one, I wonder if the authors can relate some of the MIC and growth rates observed in the evolved populations with their estimated

duration of cumulative exposure to the different antibiotics within the patients? Or if the duration of adaptation within the single patient may also play a role into the potential of the strains to be resistant to multiple drugs and to improve their growth rate.

Using treatment data of the patient from which the isolates were isolated, we tried to relate the antibiotic sensitivity of the strains analyzed with the cumulative exposure to the antibiotics of the patient. Ciprofloxacin and tobramycin are routinely used for the patient's treatment so the high MIC of the S strains might be caused by the exposure to those antibiotics. In the case of the other antibiotics, however, we were not able to relate the treatment profile with the MIC since the strains show an MIC similar to PAO1. It has been previously shown by our group that the first 2-3 years of within-patient evolution are critical for the development of antibiotic resistance toward ciprofloxacin and the decrease in growth rate (Bartell et al., 2019 doi:10.1038/s41467-019-08504-7). For strains 141 and 10 we can confirm that the duration of adaptation has a role both in the susceptibility profile and in the growth rate. We have modified the text to make clearer the treatment regime of the CF patients (lines 404-406).

Very minor point:

Fig 4 in my opinion is a bit inconsistent and redundant. For some cases there are counts and trees, for others only trees which are presented somehow differently too. This is not a big issue, but I wonder if the authors can make the figure more consistent or at least less loaded.

We have slightly modified the figure to make it more consistent. Now all the hierarchical clusters are shown similarly.

Reviewer #3 (Remarks to the Author):

This manuscript describes a laboratory evolution experiment that studies compensatory evolution in *Pseudomonas aeruginosa*. Founding strains with a slow growth phenotype were propagated in a rich environment to allow for adaptive evolution and increased growth rate. The genomes of evolved strains were sequenced to identify accumulated mutations, and differential gene expression patterns were also determined. Different founding genotypes took different adaptive trajectories during compensatory evolution. However, some of the similarities suggested that some global regulators were involved in the original slow growth phenotype, and the effects of some specific genes were confirmed by genetic analyses. In addition, the increased growth rate was correlated with a restoration of antibiotic sensitivities, suggesting a direct trade-off. Most of the methods and results are described sufficiently. Overall, the manuscript is well written and the figures nicely summarize the data and the main points. My main concerns are discussed below, along with some more specific comments on a line-by-line basis.

General concerns:

How common is reduced growth rate in CF-adapted strains? It may be a phenotypic marker of adaptation, but how important is it when compared to other phenotypic markers like mucoidy, etc?

Slow growth is one puzzling phenotype since it is still unclear if reduced growth rate is specifically selected for or if it is a cost of other more important fitness gains. In a recent publication from our group, analyzing the phenotype of 474 clinical strains of *P. aeruginosa*, we identified growth rate reduction as one of the main convergent phenotypes in adapted strains along with increased ciprofloxacin resistance and increased biofilm production. Not surprisingly such changes in phenotype occurs after only 2-3 years of adaptation, indicating a pivotal role of reduced growth rate during the infection (Bartell et al., 2019 doi:10.1038/s41467-019-08504-7). We have included this information in the introduction (lines 56-59)

More information should be given on the overall phenotypes of the founding strains and the evolved strains. For example, did phenotypes such as mucoidy, motility, biofilm formation, or pigment production change during the experiments? These are important to consider because propagation in broth alone may select for certain phenotypes, which can result in gene expression changes. For example, growth in broth selects for reduced mucoidy, so changes in alginate gene expression patterns may have arisen to that form of selection, not selection on increased growth rate per se. Similarly, growth in broth selects for reduced motility and biofilm formation, so flagella and pellicle gene expression may have been modified. So the question is: Did the laboratory evolution select for increased growth rate, or was it a by-product of selection for other factors?

We have analyzed the phenotype of the strains isolated during the ALE and we found little or not-aligned changes during evolution (Fig. S3). Moreover, we performed Pearson's correlation analysis to quantify the relationship between growth rate and motility, biofilm and pyoverdine production which showed no significant strong correlation. This indicates that, overall, our ALE conditions selected specifically for high growth rate rather than other phenotypes. We have included these new results in the text (lines 156-163).

For mucoidy, only strain 141_S showed a strong alginate production. Similar metabolic burdens causing reduced growth rate have been largely described for strains producing protein and metabolites for biotechnological applications. Even though we are unable to define if selection is on growth rate or alginate production, we think that our results shows that, in the host, alginate overproduction has a secondary effect which is reduced growth rate. This in turn helps to reduce antibiotics susceptibility, increase persistence in the patient and helps to withstand stresses and the immune system.

Why were hypermutator strains chosen? I understand that the increased rate of mutation supply can speed up the adaptive process, but many laboratory evolution experiments have been successfully performed with standard non-mutators. The problem here is that mutators can take adaptive trajectories that non-mutators may not. Not only is the mutation rate an order of magnitude greater, but mutators also have a different spectrum and directional bias of mutations. Furthermore, the greatly

increased mutation rate adds noise to the mutation patterns, so some of observed mutations may be insignificant and play no future role in adaptation. How these differences between mutators and non-mutators could impact the interpretation of the results should be discussed.

We decided to use hypermutator strains for three main reasons: from the one side hypermutator bacteria represent a large fraction of the bacterial population (30-60%) present in 30-50% of the patients; on the other side, we were concerned that ALE with normomutator strains would have required a number of generations way higher than hypermutators, therefore making it unfeasible to perform the experiment and to achieve a significant increase in growth rate. Even though hypermutator strains might take alternative adaptive trajectories from normal mutator strains, both strains develop similar phenotypes in the patients and similar mutations have also been identified. Moreover, pathoadaptive genes are shared between types of bacteria. This indicates that hypermutator strains have a pivotal role in within-patient adaptation and in the long run in the establishment of a chronic infection. We have expanded the introduction (lines 68-76) and the discussion (lines 454-468) to account for the reviewer's comments.

The results of genome sequencing and differential gene expression identified thousands of potential genes of interest. The fact that these were mutator strains definitely added to the wealth of candidate genes. However, the method used to select genes for further characterization is not clearly described. It seems that the authors focused primarily on genes or pathways that were already known to be important for *P. aeruginosa* adaptation to CF patients. The authors do state that "only some of the genetic changes correlating with an increase in growth rate will be presented" (line 188-189), but it would be useful to know how representative these discussed changes are of the total genetic changes. In other words, were these pathways objectively the most important, or were they chosen subjectively from a large number of options?

To identify potential genes involved in the reduced growth rate, we defined some criteria, presented in material and methods, that could allow us to reduce the noise caused by hypermutation and identify potential candidates in an unsupervised manner. Briefly, we excluded genes containing synonymous mutations or mutations on intergenic regions, genes already containing more than 3 non-synonymous mutation which can indicate loss-of-function, and genes unknown and uncharacterized. Using this approach, we reduced the potential candidates to a handful which could be further characterized. We discussed in the text only the most relevant in the interest of the fluidity of the text and grouped the convergent ones together to make the story less redundant. We have moved the selection criteria from material and methods to the main result section making them more accessible for the reader (lines 224-233).

A discussion of bacteriostatic versus bacteriocidal antibiotics should be presented here. Bacteriostatic antibiotics only affect cells that are actively dividing, so reduced cell division can promote cell survival. This is one explanation of how reduced growth rate can cause antibiotic resistance.

As suggested by the reviewer, we have discussed the role of reduced growth rate in relationship to antibiotic resistance for both bactericidal and bacteriostatic antibiotics (lines 498-503).

In multiple cases (see below), comparisons are made between evolved 10, 141, and 147.1 strains and the PAO1 reference genome, but this is not appropriate and can add confusion the interpretation of the results. Only the evolved PAO1 strains should use the PAO1 reference genome as the ancestor reference. For evolved 10, 141, and 147.1 strains, the 10S, 141_S, and 147.1_S genomes should be used as the ancestor reference, respectively.

We compare our strains to PAO1 only in the case of the S clinical strains. For the I and L strains, instead, we only use each corresponding S strain. We use PAO1 as reference to identify potential mutations in the S strains genome which otherwise could not be identified. For example, in the case of *topA* mutations, we would not have been able to identify the mutation in *gyrA* if we would have only used strain 10_S as reference. Similarly, for *rpoN* mutations, if we would have used the S strain, we would have just considered the 2 mutations accumulated in the L strains as new mutation instead of compensatory mutations for the nonsense mutation in the S strain. We are aware that the comparison can create confusion, but we think that it is necessary for our analysis. Moreover, in many manuscript PAO1 is used as reference for comparative analysis and to identify mutations in clinical isolates (Marvig et al., 2015 doi:10.1038/ng.3148). To avoid confusion, we have explained more in detail our comparative approach in the text (lines 170-175).

Specific comments (line number):

102: Is growth rate in “cell doublings per hour”?

Yes, it is. We have included the information in the text (line 126).

105: The graph in Fig 1a shows that the S version of PAO1 had a slower growth rate than the I or L versions.

Even though the growth rate of PAO1_S looks lower than it should, that depends on the fact that the growth rate is automatically calculated using our in-house MATLAB script which forecast when during the ALE another sample for OD₆₀₀ measurement should be take. Sometimes the prediction is not perfect and the sample is taken in stationary phase, so the growth rate is miscalculated. For this reason, for each strain analyzed in detail in this study (WGS, RNA-seq, phenomics and MIC), we have calculated the growth rate using an exponential fitting of the OD₆₀₀ data using a conventional method as indicated in material and methods and presented the data in Fig. S3.

129-132: This concept is known as the contingency of evolution.

We have changed “strain specific evolutionary histories” in “historical contingency” (line 165).

139-140: Each founder S strain is different from each other, and different from PAO1, so the number of mutations between an S strain and PAO1 is not relevant at this point. I would remove this comparison, and remove these data from Figure 2a to avoid confusion. Also, I would call these differences “SNPs”, not “mutations”. The important comparison is between the founder S strain and the evolved I and L strains.

We understand that the information can create confusion, but we think that the information is relevant. The goal of our work is to analyze 1) the evolution of the I and L strains during ALE, 2) track back the evolution history of the starting strains in the patients. Without the comparison to PAO1 we wouldn't be able to identify important mutation in the S strains (for example the *gyrA* and *rpoN* mutation in the 10_S strain or the mutations on the ribosomal proteins in the 427.1_S strain) and to infer the trajectories of evolution in the patients. We prefer keeping the information so the readers can have a complete overview of the mutations present in the starting strains and accumulated during ALE. To avoid confusion, we have explained more in detail our comparative analysis in the text (lines 170-175). In Fig. 2a, we have modified “mutations” in “SNPs/Indels”.

154-156: I have multiple points for tree in Figure 2c.

1) Why was maximum parsimony used, rather than maximum likelihood?

2) The phylogenetic tree in Figure 2c assumes the terminal taxa are contemporaries, but that is not the case because S evolved into I, and I into L. So can this be displayed in a different way, one that reflects these relationships?

3) If a tree is still used, then a separate tree should be constructed for each strain group, with the S version as the root. Again, the relationships between the founding strains (and PAO1) are not relevant and will only add confusion to the interpretation.

4) Why are the expected relationships within strain groups not recovered? For example, if each replicate in each strain group is evolving independently, I1 should be more closely related to L1, and I2 to L2, and I3 to L3, correct? That does not appear to be the case. Also, since number of generations of evolution should be correlated with number of accumulated mutations, all I (intermediate) isolates should have less mutations than L isolates, but that does not appear to be the case either.

We used maximum parsimony to minimize the number of changes (mutations) required to explain the increase in growth rate. However, a similar result is obtained when using maximum likelihood. We have now included a new supplementary figure with both trees (Fig. S4a).

We think that a tree is the most straight forward representation of the relationship between the strains. Therefore, as indicated by the reviewer, we have created for each group of strains single trees using both maximum parsimony and maximum likelihood rooted on the S strain. The trees are shown in Figure S4b.

Even though the relationship between the replicates should be maintained, we can't be sure that in the population from which we isolated the strains, we picked the strain directly related to the previous one. Due to hypermutation, the variability between strains within the population might be extremely high for those mutations without fitness cost. Moreover, mutations can also be reverted or lost during a certain time of

evolution to reappear again in a following population. Importantly, mutations leading to increased growth rate are maintained in the corresponding population (see the case of strains 10) indicating that the relationship between strain for the high impact mutations is maintained.

Except for the case of the intermediate strains 141_I1 and 10_I1 which harbor more mutations than the corresponding late strains 141_L1 and 10_L1, in the rest of the pairs the number of mutations follow the number of generations, therefore increasing over time.

167-169: Based on Figure 2d, only Strain 10 evolved towards the area occupied by PAO1. The other 2 strains are distinct in PCA-space, so why was Strain 141 said to evolve towards PAO1 as well?

In the text we want to underline the distinct trajectories on PC1 of strains originated from 141_S and 10_S relative to those originated from 427.1_S. The sentence, indeed, should be read together with the following one (lines 206-211). Even though strains 141 never occupy the area of PAO1 strains both in the whole genome and in the core genome they move toward the area of PAO1 which represent the optimum.

201-204: Were there any mutations within the *algU* and *mucA* genes themselves?

No mutations were accumulated during ALE in the genome of 141 strains. We included this information in the text (lines 246-249).

252-255: Much like the *topA* mutations, the effects of the *ntrC* mutations are highly speculative and could use some more genetic support.

We agree with the reviewer that the effect of the *ntrC* mutation is speculative, but we prefer focusing in the paper on those mutations (*gyrA*, *topA* and *rpoN*) which were identified in more than one population. However, it is a confirmation of the importance of *rpoN* gene to find a mutation in *ntrC* which is part of the *rpoN* regulon.

265: Recent studies with *P. aeruginosa* have found that ribosomes and DNA replication machinery are under selection in isolates from CF patients.

We agree with the reviewer and indeed our group published in 2019 a paper on the characterization of a ribosomal mutation in *P. aeruginosa* causing high resistance to aminoglycosides and a severely reduced growth rate (Halfon et al., 2019 doi: 10.1073/pnas.1909831116). We have included the reference in the text (lines 329-334).

317: In Figure 4g, I assume the asterisks indicate SNPs between S strains and reference PAO1. Again, this comparison should be removed because PAO1 is not the ancestor of the S strains, and this can confuse the interpretation of the comparison between S and ALE.

Yes, the asterisks indicate SNPs present in the starting strains relative to PAO1, therefore accumulated during within-patient evolution. We think that the information is relevant because allows to show the presence of high selective pressures on virulence factors in CF in comparison to the ALE condition.

333: On average, the intrinsic sensitivity of PAO1 and 141 is not much higher than for strain 10. The trend is really being driven by the high resistance strain 427.1.

We have changed the focus of the paragraph to underline the changes in antibiotic susceptibility independently of the clinical break-points of the antibiotic and, therefore, the definition of resistant/sensitive bacteria (lines 397-442).

337-340: Interesting to find a lack of mutations in well-know resistance genes, suggesting the compensatory mutations were located in other unknown genes.

Although unlikely, we can't completely exclude that mutations in unknown genes are the cause on the reduced antibiotic susceptibility. To expand our analysis, we looked at the transcriptional profiles of known antibiotic resistance genes and found that only in few cases we could associate the change in expression profile to the increased susceptibility (lines 400-425).

407-408: I would be careful with this statement. Just because the increase in growth rate took different trajectories, it doesn't mean the original decrease in growth rate took different trajectories. In theory, the same genotype can follow different adaptive strategies based on mutation supply, and the likelihood of this would increase in these mutator strains.

We agree with the reviewer that it is not easy to define the trajectory of evolution which in the first place caused the decrease in growth rate. However, in the sentence we wanted to underline that the different strains 141, 10 and 427.1 showed different mechanisms of increased growth rate, therefore suggesting that at least 3 different trajectories exist. We toned down the sentence changing "confirms" to "suggests" (line 514).

452-454: This may be an over-statement of the confidence in the conclusion. These experiments can suggest which mechanisms caused the slow growth rate, but they cannot determine if the slow growth rate was the target of selection, or if it is a by-product of selection on another phenotype.

We agree with the reviewer and we have included this open question in the text (lines 560-562).

518: Gene expression patterns were determined for growth in aerated LB broth. Studies have demonstrated that gene expression patterns in a medium that more closely simulate the CF lung environment may be quite different from that in rich LB medium. It is possible that the slow growth phenotype interacts specifically with characteristics of the CF lung environment, and these may not be captured in LB. The authors may want to discuss this as a potential limitation.

We agree with the reviewer that growth rate reduction might be also a function of the gene expression profile in the CF environment which is different from the LB condition used in this study. However, only using a medium different from the one present in the lungs of CF patients, we could force evolution toward new fitness peaks. Otherwise, bacteria would have maintained their mutational and

transcriptional profile due to the similarities between the conditions. Therefore, in light of our goal, we think that this is not a limitation.

546-551: This paragraph highlights one of the issues with using hypermutable strains for adaptive evolution experiments. The greatly increased mutation rate adds so much noise to the mutation patterns, so that the importance of the observed mutations cannot be determined. Newly generated mutations may not have been exposed to selection yet, so they can be observed even though they will not play a role in adaptation. To compensate, the authors end up filtering out and ignoring a large proportion of the mutations. So although some evolved 141 strains have over 2800 mutations, the focused on 17 genes from these evolved lines.

We agree with the reviewer that hypermutation adds noise to the mutation pattern. However, hypermutator strains represent a big portion of the infecting bacteria pushing forward evolution in the CF environment. Even though we filtered out most of the mutations, we were able to identify specific mechanism and mutations causing increased growth rate as validated in our molecular analysis.

924: Figure S3 panels b,c,d - Again, the comparison of strains back to reference PAO1 should be removed. For panel a, this comparison makes sense because PAO1 is the ancestor of the evolved strains, and there are no SNPs at generation 0. For panels b,c,d, the reference ancestor should be the respective S strain, and there should also be no SNPs generation 0.

As indicated in the above comment, the goal of the work is to show the evolution of the I and L strains during ALE, and to track back the evolution history of the starting strains in the patients. Therefore, we think that the information is relevant for reader.

REVIEWERS' COMMENTS

Reviewer #1 (Remarks to the Author):

The authors have provided a thorough and reasonable response to all comments. Where necessary, critical revisions and additional analyses have been added. My initial concerns have been adequately addressed. I believe that the revised manuscript provides important insight into the evolutionary trajectories of chronically adapted *P. aeruginosa* clinical isolates and will be of interest to the broader scientific audience.

Reviewer #2 (Remarks to the Author):

The authors have covered all of my concerns and I have no additional comments. I think the paper adds important insights into the field and will be happy to see it published.

Reviewer #3 (Remarks to the Author):

This revised manuscript is greatly improved and the authors have made significant efforts to address the concerns of the original reviews. In general, the revisions have satisfied the issues I originally noted.

Remaining comments:

Line 99: The authors state that "Our analysis reveals that the slow growth phenotype may be achieved in steps.", but it should be "the reversion of the slow growth phenotype".

Line 111-12: The author's state that "only growth rate based selection" was applied, but that is not necessarily true. The methods of propagation used during ALE selected for higher fitness, and growth rates are just one component of fitness.

The Supplementary Figures will need to be re-numbered based on order of appearance.

For Supp Fig 10, panels (a) and (b) are switched.

General comment: To make inferences about the evolution from a naïve to adapted state, the authors use "reverse evolution" to study the changes from adapted back to the naïve state. There are few caveats to this approach that should be made more clear in the discussion.

First, the environment selecting for the "naïve" state in this ALE (rich laboratory medium) is not the same as the original "naïve" environment (soil, water, animal, acute infection?). So evolution back to fast growth in the lab is not really a reversion, because it will NOT be back-tracking along the same genetic route that caused reduced growth during the original naïve-to-adapted transition. Since growth rate is such a complex phenotype, one could consider the rich laboratory medium as a completely novel environment, with no expectations of "reversion".

Second, even if it was evolution back to the original naïve state, the same reverse route could be taken by evolution of increased growth rate, but many other routes are also possible. In reality, there is a certain amount of stochasticity, and trajectories taken during evolution of slow growth may be drastically different than the trajectories taken during evolution of fast growth. In many cases, they are related, but this is not guaranteed. The point is: Discovering the genetic mechanism for growth rate recovery does not mean it was the original mechanism for growth rate reduction. It is suggestive, but not definitive, and the authors should make this clear in their concluding statements.

Overall, this is a very comprehensive study addressing an important scientific question. The

manuscript is well-written and the figures represent the finding very nicely.

REVIEWERS' COMMENTS

Reviewer #1 (Remarks to the Author):

The authors have provided a thorough and reasonable response to all comments. Where necessary, critical revisions and additional analyses have been added. My initial concerns have been adequately addressed. I believe that the revised manuscript provides important insight into the evolutionary trajectories of chronically adapted *P. aeruginosa* clinical isolates and will be of interest to the broader scientific audience.

We thank the reviewer for the comment and appreciation.

Reviewer #2 (Remarks to the Author):

The authors have covered all of my concerns and I have no additional comments. I think the paper adds important insights into the field and will be happy to see it published.

We thank the reviewer for the comment and appreciation.

Reviewer #3 (Remarks to the Author):

This revised manuscript is greatly improved and the authors have made significant efforts to address the concerns of the original reviews. In general, the revisions have satisfied the issues I originally noted.

Remaining comments:

Line 99: The authors state that “Our analysis reveals that the slow growth phenotype may be achieved in steps.”, but it should be “the reversion of the slow growth phenotype”.

Changed as requested.

Line 111-12: The author's state that “only growth rate based selection” was applied, but that is not necessarily true. The methods of propagation used during ALE selected for higher fitness, and growth rates are just one component of fitness.

We changed “only growth rate based selection” to “in absence of selective pressures”.

The Supplementary Figures will need to be re-numbered based on order of appearance.

We have re-numbered the supplementary figures as requested.

For Supp Fig 10, panels (a) and (b) are switched.

We have corrected the panels.

General comment: To make inferences about the evolution from a naïve to adapted state, the authors use “reverse evolution” to study the changes from adapted back to the naïve state. There are few caveats to this approach that should be made more clear in the discussion.

First, the environment selecting for the “naïve” state in this ALE (rich laboratory medium) is not the same as the original “naïve” environment (soil, water, animal, acute infection?). So evolution back to fast growth in the lab is not really a reversion, because it will NOT be back-tracking along the same genetic route that caused reduced growth during the original naïve-to-adapted transition. Since growth rate is such a complex phenotype, one could consider the rich laboratory medium as a completely novel environment, with no expectations of “reversion”.

We agree with the reviewer that evolving adapted clinical isolates in a new environment doesn't imply to revert them to a naïve state. But we use the term “reversion” since we focus on the compensatory events which brought slow growing bacteria to the high growth rate of the ancestors. We have presented this concept in the discussion (lines 593-595).

We have changed in the text all the sentences where reversion back the naïve state was indicated.

Second, even if it was evolution back to the original naïve state, the same reverse route could be taken by evolution of increased growth rate, but many other routes are also possible. In reality, there is a certain amount of stochasticity, and trajectories taken during evolution of slow growth may be drastically different than the trajectories taken during evolution of fast growth. In many cases, they are related, but this is not guaranteed. The point is: Discovering the genetic mechanism for growth rate recovery does not mean it was the original mechanism for growth rate reduction. It is suggestive, but not definitive, and the authors should make this clear in their concluding statements.

We have included this information in the discussion (lines 622-626)

Overall, this is a very comprehensive study addressing an important scientific question. The manuscript is well-written and the figures represent the finding very nicely.

We thank the reviewer for the comment and appreciation.